# Computational mechanisms for temporal integration in the anterior claustrum

**Kuenbae Sohn[†], Donghyeon Yoon[†], Junghwa Lee\*, Sukwoo Choi\***

School of Biological Sciences, College of Natural Sciences, Seoul National University, Seoul, Republic of Korea

## eLife Assessment

This work provides an **important** modeling-based framework for understanding the processes of temporal integration in the claustrum. These mechanisms could support a broader range of integrative brain function. The manuscript presents **solid** evidence for how claustrum may integrate temporal disparate signals via a novel computational phenomenon with neural dynamics evolving along neural trajectories as opposed to settling into fixed-point attractor states.

**\*For correspondence:**
antares0715@gmail.com (JL);
sukwoo12@snu.ac.kr (SC)

[†]These authors contributed equally to this work

**Competing interest:** The authors declare that no competing interests exist.

**Abstract** The claustrum, with its extensive reciprocal connections to nearly all cortical regions, has long been hypothesized as a key hub for integrating diverse cognitive, sensory and motor information. However, despite its anatomical connectivity, whether and how it functionally integrates different inputs to generate coherent representations has remained unclear. Here, we developed a recurrent neural network (RNN) trained via supervised learning on behavioral metrics of delayed escape—a behavioral paradigm that requires integration of temporally separated task-relevant signals. A subset of RNN neurons exhibited dynamics similar to those of anterior claustral neurons during this behavior. These neurons formed a recurrent cluster, a structure supported by in vitro stimulation experiments in claustral brain slices. We analyzed the computational properties of this claustrum-like cluster via dimensionality reduction of population activity. The network showed nonlinear integration of temporally distributed inputs and increased synergistic information. Rather than settling into attractors, integrated information was dynamically encoded along continuously evolving neural trajectories. Notably, similar trajectory patterns associated with dynamic integration were observed in claustral recordings, suggesting the model's biological plausibility. We propose that the anterior claustrum dynamically integrates task-relevant input signals over time and broadcasts the evolving representation to downstream brain regions capable of reading and interpreting it in a context-dependent manner.

## Introduction

The claustrum is a thin, elongated sheet of gray matter that maintains extensive reciprocal connections with nearly all cortical areas and several subcortical regions. This dense interconnectivity has led to the longstanding hypothesis that the claustrum serves as a hub for integrating internal and external signals into a unified percept, potentially supporting conscious awareness and higher-order cognition (*Crick and Koch, 2005*). However, direct experimental evidence for this hypothesis has remained scarce. One recent study reported that individual claustral neurons can receive convergent input from multiple cortical areas, which is critical for a behavioral task requiring multimodal sensory inputs (*Shelton et al., 2025*). In contrast, numerous studies, including those from the Mathur and Citri groups, have implicated the claustrum in diverse cognitive processes such as attention, salience detection, and cognitive control (*White et al., 2018*; *Atlan et al., 2024*; *Faig et al., 2024*; *Niu et al.,*

*2022*; *Chevée et al., 2022*; *Narikiyo et al., 2020*; *White et al., 2020*; *Qadir et al., 2022*; *Goll et al., 2015*; *Atlan et al., 2018*; *Terem et al., 2020*; *Madden et al., 2025*). This apparent discrepancy may reflect differences in the specific claustral subregions examined. Although the claustrum is less differentiated in rodents than in primates, important distinctions exist even among rodent species (*Chong and Gămănuţ, 2024*). In mice, the anterior boundary of the claustrum typically does not extend beyond the rostral end of the striatum (*Wang et al., 2017*). In contrast, in rats, the claustrum extends further rostrally beneath the forceps minor of the corpus callosum (fmi), where claustrum-specific gene expression has been reported (*Han et al., 2024*; *Dillingham et al., 2019*).

Only a few previous studies have directly investigated this rostral segment of the rat claustrum (rostral-to-striatum, rsCla) (*Dillingham et al., 2019*; *Zhang et al., 2001*; *Jankowski and O'Mara, 2015*; *Jankowski et al., 2017*; *Grasby and Talk, 2013*). However, recent findings suggest that neuronal activity in this region plays a critical role in behavioral tasks in which two different kinds of information must be integrated (*Han et al., 2024*; *Park et al., 2025*), indicating a potentially distinct computational role for the rsCla. Among these tasks, the newly developed delayed escape task provides a behavioral paradigm in which rats must rapidly escape to a neutral zone following a fear-inducing conditioned stimulus (CS), after a delay that separates the CS from the opening of an outlet. Crucially, the task requires no prior training and instead depends on the flexible integration of two temporally separated events that become jointly relevant for guiding escape behavior. Neural recordings from this task revealed that CS-related information was maintained through persistent activity in rsCla neurons, suggesting that these neurons may integrate the sustained internal threat signal with the subsequent outlet-opening signal to guide future escape behavior.

To investigate how the anterior claustrum integrates information at the population level, we employed a delayed escape task in Long-Evans rats. Because the task could only be performed once per animal, and only a small number of single units were recorded per subject, direct population-level analysis was severely limited. To overcome these constraints, we trained a recurrent neural network (RNN) model solely on behavioral outputs—a strategy previously shown to yield biologically plausible dynamics (*Ehrlich et al., 2021*; *Ehrlich and Murray, 2022*; *Kim and Sejnowski, 2021*; *Stroud et al., 2023*; *Brody et al., 2003b*; *Chaisangmongkon et al., 2017*; *Mohan et al., 2021*). We then compared the network's emergent population activity patterns with in vivo claustral recordings, and used the model to explore circuit-level mechanisms of integrating temporally separated inputs.

## Results
### RNNs trained on the delayed escape task show claustrum-like dynamical patterns

The delayed escape task allows only a single test trial per animal, which means that opportunities to obtain neurophysiological data are extremely limited. To overcome this limitation, we used RNN simulations, a strategy that has been widely adopted in recent neuroscience research (*Kim and Sejnowski, 2021*; *Stroud et al., 2023*; *Stroud et al., 2024*). In the delayed escape task, when a CS previously associated with electric shock is presented for the first time in a novel environment, animals infer that a value-neutral alternative space is likely to be safer and therefore exhibit faster escape behavior (*Figure 1A*, This figure is adapted from Figure 1A of *Han et al., 2024*). Importantly, the passage to this neutral space opens after a delay following CS offset, making it essential for CS-related information to be maintained beyond the actual presence of the CS. As a control group, animals were placed in the same environment during the CS period but did not receive the CS. In previous experiments, animals that received the CS escaped to the neutral space significantly faster than controls (*Han et al., 2024*). To model this behavior, we built upon the continuous-time RNN framework developed by *Ehrlich et al., 2021* and *Ehrlich and Murray, 2022*; *Figure 1B*.

The trained RNN reproduced the escape latency pattern reported by *Han et al., 2024*: the CS + door-opening condition escaped faster than the control door-opening only group (*Figure 1C*), reflecting the in vivo behavioral results. Furthermore, it showed similarities to the claustral network dynamics observed in that study (*Figure 1D–M, D–G and L (left), and M (left)* are adapted from Figure 4B, D, E and S4E of *Han et al., 2024*). In the previous biological study, the responses of individual claustral units representing each behavioral epoch were subjected to dimensionality reduction and clustering (*Figure 1D*). One of these clusters showed persistent firing during and after CS presentation, and the

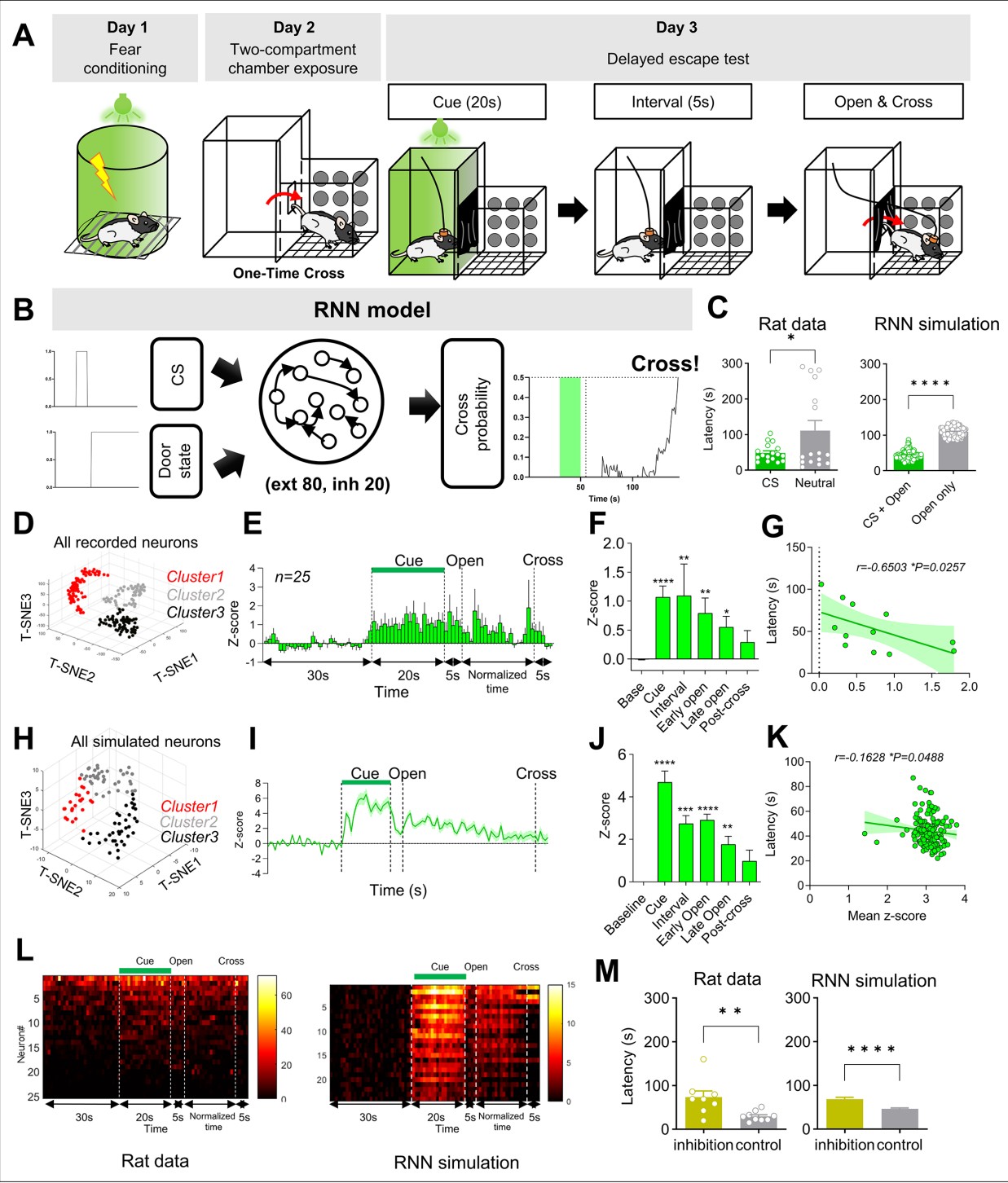

**Figure 1.** RNN simulation capable of performing the delayed escape test. (**A**) Schematic diagram of the delayed escape task. On day 1, animals were conditioned with a light cue paired with an electric footshock. On day 2, they were adapted to the test context with only a single crossing permitted. On day 3, animals were placed in one compartment of the test context and presented with a 20-s CS, followed by a 5-s delay, after which the outlet door was opened to allow escape. The test was conducted only once. (**B**) Schematic diagram of the task structure that the RNN model was designed to simulate. (**C**) Graphs of crossing latency, measured as the interval from door opening to crossing in rats and as derived from RNN simulations. (Rats: CS, n = 17; Neutral, n = 16; *p = 0.0335, unpaired *t*-test; RNN: CS, *n* = 147; Open, *n* = 108; ****p < 0.0001, Mann–Whitney). (**D**) Clustering of all recorded claustral neurons in rats (*n* = 203). (**E**) Mean *Z*-scored firing rates of non-exploratory Cluster 1 neurons in the CS group (*n* = 25, rat data). Non-exploratory Cluster 1 denotes the cluster showing increased persistent activity both during CS presentation and thereafter. The CS group corresponds to the CS +door-opening condition in the RNN model. (**F**) Average Z-score for each behavioral period shown in E. Bars: Friedman test, ****p < 0.0001; Dunn's

*Figure 1 continued on next page*

*Figure 1 continued*

multiple comparisons—Base vs Cue, ****p < 0.0001; Base vs Interval, **p = 0.0085; Base vs Early Open, **p = 0.0075; Base vs Late Open, *p = 0.0455; Base vs Post-cross, p > 0.9999. (**G**) Correlation between crossing latency and Cluster 1 activity in the CS group (n = 12; Spearman r = 0.6503, *p = 0.0257). (**H**) Clustering for simulated neurons (n = 100 neurons). (**I**) Mean Z-scored firing rates of Cluster 1 neurons in the CS +door-opening condition (n = 24 neurons, RNN) across all simulated trials. (**J**) Average Z-score for each behavioral period shown in I. Bars: Friedman ****p < 0.0001; Dunn's— Base vs Cue, ****p < 0.0001; Base vs Interval, ***p = 0.0001; Base vs Early Open, ****p < 0.0001; Base vs Late Open, **p = 0.0034; Base vs Post-cross, p = 0.7134. (**K**) Correlation between crossing latency and Cluster 1 activity in the CS + door-opening condition (147 trials; Spearman r = –0.1628, *p = 0.0488). (**L**) Heatmaps of single-neuron activity corresponding to panels E and I. Left: rat data (firing rate, Hz); Right: RNN data (Z-score). Neurons are ordered by overall activity. (**M**) Crossing latency in rats with anterior claustrum inhibition during the 5-s period between CS offset and door opening, compared with control virus-expressing animals (left, inhibition, n = 8; control, n = 9; **p = 0.0079, Mann–Whitney). Corresponding results in the RNN are shown for selective inhibition of Cluster 1 neurons during the same 5-s window simulation (right, inhibition, n = 28; control, n = 37; ****p < 0.0001, Mann–Whitney). Bars show mean ± SEM. Panels A, D–G, and L (left) are adapted from Figures 1A and 4B, D, E, and S4E of *Han et al., 2024*, *Cell Reports*, Elsevier.

The online version of this article includes the following figure supplement(s) for figure 1:

**Figure supplement 1.** RNN activity of Clusters 1–3 under CS and door-opening only conditions.

**Figure supplement 2.** Contribution of each cluster to the output during the delayed escape test.

**Figure supplement 3.** RNN activity of Clusters 2 and 3 under inhibition and escape latency under 180-s delay condition.

magnitude of this increase was inversely correlated with escape latency (*Figure 1E–G*). From this, we concluded that this cluster maintained CS signals until the door to the escape path opened, thereby contributing to delayed escape.

To determine whether this cluster was reproduced in the trained RNN, unit responses in the RNN representing each behavioral epoch were also subjected to dimensionality reduction and clustering. One of the clusters displayed the same persistent activation during and after CS presentation as observed in the in vivo claustrum (*Figure 1H–J*). As in the biological in vivo data, stronger activation in this model cluster predicted shorter escape latency (*Figure 1K*). Single-unit responses were also similar between real and simulated data: the persistence after the CS did not arise from a single unit firing continuously but rather from many units intermittently increasing their firing, which overlapped to form a persistent signal at the population level (*Figure 1L*).

Even when no CS was presented and only the door opening cue was given (the door-opening condition), the population activity pattern generated by the RNN showed a pattern of gradually increasing after the door opened and then decreasing as the crossing approached (*Figure 1—figure supplement 1A*). The other two model clusters exhibited different patterns: Cluster 2 was generally suppressed regardless of CS presence, and Cluster 3 gradually ramped up from the door opening cue to just before the crossing, also regardless of CS presence (*Figure 1—figure supplement 1B–E*).

Next, we quantified the contributions of the three clusters to the RNN output when the CS was presented. The claustrum-like Cluster 1 contributed very little to the output, whereas Cluster 3 was dominant in influencing cross behavior. This suggests that Cluster 1 may not directly drive the output but instead processes the CS and door opening signals and broadcasts this information to other clusters (*Figure 1—figure supplement 2*). In our previous study (*Han et al., 2024*), optogenetic inhibition of the anterior claustrum during the delay period between CS offset and door opening significantly increased escape latency (*Figure 1M*, *Figure 1—figure supplement 3A*). Introducing inhibitory input to suppress RNN Cluster 1 during the same period produced a similar increase in latency (*Figure 1M*), whereas inhibiting the other clusters had no effect (*Figure 1—figure supplement 3B*). Furthermore, when the interval between CS and door opening was extended from 5 s to 180 s in vivo (*Han et al., 2024*), the latency difference between the CS and neutral groups disappeared. Indeed, the RNN also reproduced this result under the same extended interval (*Figure 1—figure supplement 3C*).

Taken together, the claustrum-like Cluster 1 in the RNN reproduced the firing pattern and behavioral correlation of the persistent activity cluster observed in vivo and resembled several key characteristics of actual claustral neurons in almost every respect, including latency increases following inhibition. This trained RNN model therefore enables deeper analyses under experimental conditions that are otherwise difficult to implement.

## Claustral neurons exhibit recurrent connectivity similar to that of the RNN

Because an RNN, by definition, contains recurrent connections among its units, finding a claustrum-like cluster whose properties mirror those of the real claustrum suggests that biological claustral neurons may likewise be interconnected through recurrent circuitry. As expected, the RNN exhibited strong recurrent weights within Cluster 1 (*Figure 2A*). If a similar architecture exists in vivo, the first prediction is that excitatory connectivity should be detectable between claustral neurons.

To test this, we sparsely expressed ChrimsonR in a subset of claustral cells and prepared sagittal brain slices for patch-clamp recordings to measure synaptic currents (*Figure 2B, C*). We activated ChrimsonR-expressing presynaptic neurons with 617 nm light, while patching ChrimsonR-negative neurons to isolate synaptic responses. Patching ChrimsonR-positive cells would have risked contamination by direct photocurrents in addition to synaptic responses. Wide-field illumination induced excitatory postsynaptic currents (EPSCs) that were abolished by NBQX + D-AP5, confirming that they were mediated by glutamatergic transmission (*Figure 2D–E*). One remaining concern was that these EPSCs might have originated from ChrimsonR-expressing axons projecting into the claustrum from outside regions. To rule this out, we employed high-resolution optical mapping. The wide field was divided into a 30 × 30 grid (900 pixels), and each pixel was stimulated for 2 ms (*Figure 2F*). EPSCs could be elicited not only near the recorded soma but also from distant pixels, indicating the presence of local excitatory-to-excitatory connectivity within the claustrum itself (*Figure 2G, H*). These findings are consistent with the previous study by *Orman, 2015* and *Shelton et al., 2025* but contrast with reports that the posterior claustrum exhibits little excitatory-to-excitatory coupling (*Kim et al., 2016*). In particular, *Orman, 2015* reported that persistent increases in claustral activity were observed only when brain slices were prepared at a specific slicing angle.

A more stringent prediction of recurrent wiring is that a brief, strong stimulus should induce sustained neuronal activity. To test this, we expressed the calcium sensor GCaMP6f in the anterior claustrum and delivered a 1-s train of electrical stimulation (20 Hz, 200 µA) in horizontal slices (*Figure 2I–K*, *Figure 2—figure supplement 1C*). This stimulation evoked calcium signals that remained elevated for over 10 s. Consistently, patch-clamp recordings from GCaMP6f-positive cells confirmed a prolonged increase in firing rate over a similar period (*Figure 2—figure supplement 1B*). Notably, coronal slices failed to show such persistence, suggesting that horizontal slicing better preserves recurrent excitatory circuits. To determine whether this persistent activity depends on synaptic reverberation, we locally pressure-injected NBQX + D-AP5 at the stimulation site during the sustained phase of the calcium signal. As predicted, the persistent response collapsed rapidly (*Figure 2L, M*, *Figure 2—figure supplement 1D–F*), indicating that excitatory synaptic transmission within the claustrum is essential for generating the sustained activity.

The RNN reproduced a similar phenomenon (*Figure 2N*, *Figure 2—figure supplement 1F*). Isolating Cluster 1 from the trained network, we reduced all inhibitory weights by 60%. When a 1-s CS-strength input was delivered, the cluster exhibited >10 s of persistent activity, matching the slice data. Reducing inhibitory weights facilitates the persistent activity, which may reflect the reduced inhibitory connectivity often present in horizontal slice preparations. Next, reducing 60% of excitatory weights for only 10% of Cluster 1 units—mimicking the spatially restricted application of NBQX and D-AP5—largely abolished the persistence (*Figure 2N*, *Figure 2—figure supplement 1D–F*). Together, these slice and RNN experiments suggest that the anterior claustrum contains recurrent excitatory connections capable of supporting prolonged activity.

## Principal component analysis-based trajectory analysis characterizes RNN dynamics

Having established that claustral neurons exhibit recurrent connectivity capable of sustaining activity, we next examined how such dynamics manifest at the population level in the RNN. Based on the results so far, the claustrum-like cluster in the RNN resembles the cluster showing persistent activity after CS presentation in the anterior claustrum. To visualize the population activity dynamics of the RNN clusters—particularly the claustrum-like cluster— we applied principal component analysis (PCA) to Z-scored firing rates from three conditions: CS + door-opening, door-opening only, and CS only (*Figure 3A–C*, *Figure 3—figure supplement 1*). At each time point, the ensemble firing rates across all neurons in each cluster were treated as a population activity vector and projected into a

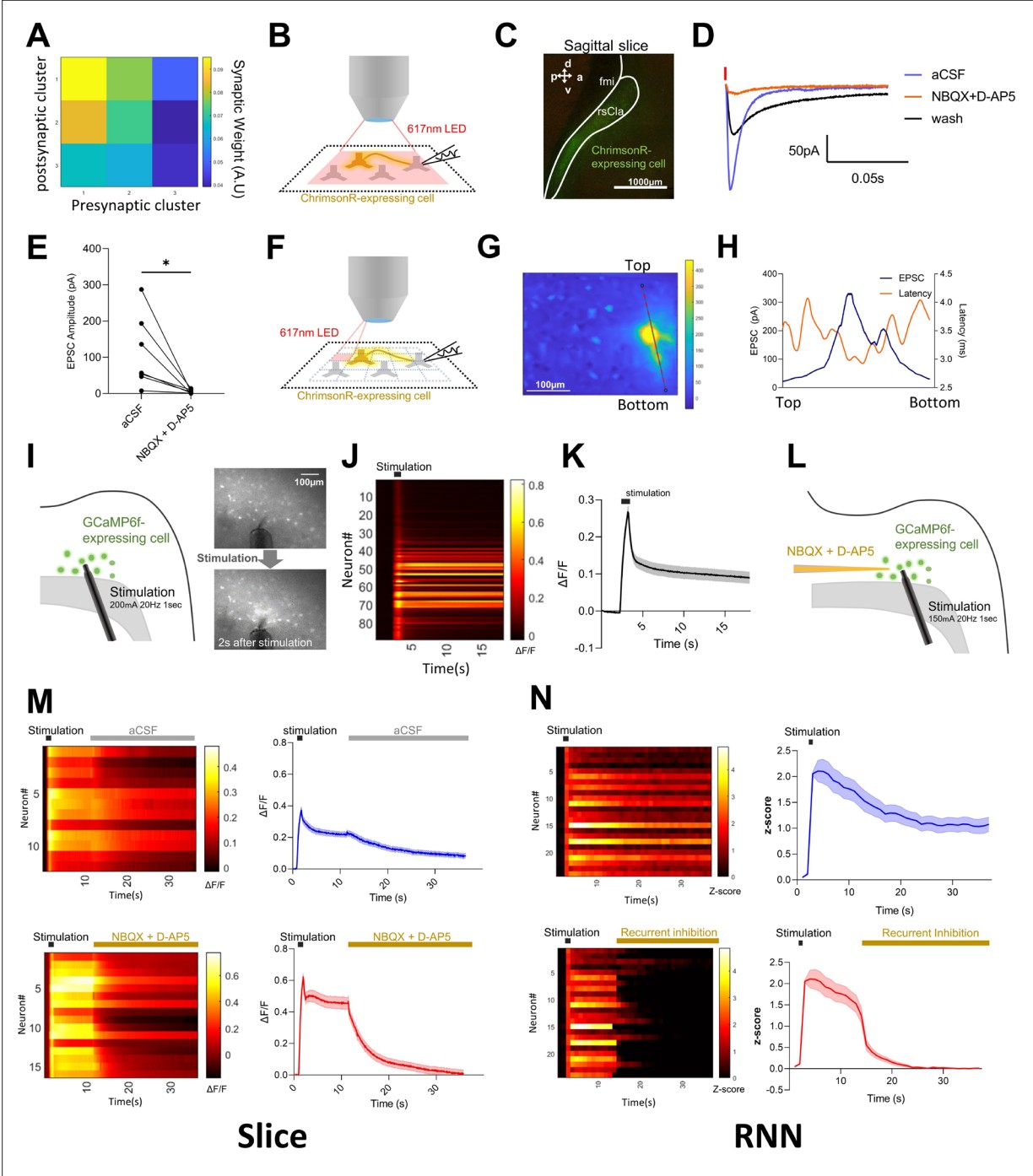

**Slice**

**RNN**

**Figure 2.** Comparison of recurrent connectivity in the claustrum and in the RNN simulation. (**A**) Mean absolute inter-cluster synaptic weight in the RNN. (**B**) Schematic representation of whole-cell recordings of ChrimsonR-non-expressing neurons during optogenetic stimulation using LEDs to stimulate ChrimsonR-expressing neurons across the entire optical field. (**C**) Confocal image showing the expression pattern of the ChrimsonR virus in sagittal claustrum slices. (**D**) Representative EPSCs evoked by 2 ms LED pulses before and after pharmacological treatments (red bar indicates stimulation). (**E**) Pooled data of EPSC amplitudes (*n* = 7) with statistical analysis using the Wilcoxon matched-pairs signed rank test (*p = 0.0156). (**F**) Schematic depiction of local stimulation using a digital mirror device (DMD). (**G**) Optical stimulation of each divided part produces variable amplitudes of EPSCs in whole-cell patched neurons expressing no ChrimsonR. Heatmap showing EPSC amplitudes upon stimulation of the designated part of optical field. Please note that when stimulating closer to the recorded neuron, larger amplitude EPSCs were induced. (**H**) The graph displays EPSC amplitudes and latencies in the region marked with dashed lines on the heatmap shown in G. (**I**) Left: schematic representation showing an experimental configuration in which brief electrical stimulation produces a persistently enhanced activity in rsCla slices. Right: representative images of GCaMP6f fluorescence changes immediately before and 2 s after stimulation. (**J**) Heatmap of fluorescence changes for individual puncta in the slice shown in I. (**K**) Population-

*Figure 2 continued on next page*

*Figure 2 continued*

averaged fluorescence trace of all puncta in J (*n*=89; mean ± SEM). (**L**) Schematic representation showing an experimental configuration in which effects of the blockers for AMPA/NMDA receptors on the persistent response were examined. (**M**) Calcium imaging before and after an aCSF puff (top panels; *n* = 12) and an NBQX + D-AP5 puff (bottom panels; *n* = 16): heatmaps (left) and population-averaged traces (right). (**N**) RNN analogue: heatmaps (left) and averaged *Z*-scores (right) for Cluster 1 neurons following brief excitation (top panels; *n* = 24) and after the addition of recurrent inhibition (bottom panels; *n* = 24). Shaded areas represent SEM.

The online version of this article includes the following figure supplement(s) for figure 2:

**Figure supplement 1.** Slice physiology and pharmacological manipulation of claustral persistent activity.

three-dimensional space defined by the top 3 principal components. Because cross-latency (the time from door opening to actual crossing) varied across trials, the interval from door opening to crossing was time-normalized within each trial. For the CS only condition, in which cross-latency was undefined, we used the mean cross-latency from the CS + door-opening condition to define a notional crossing time.

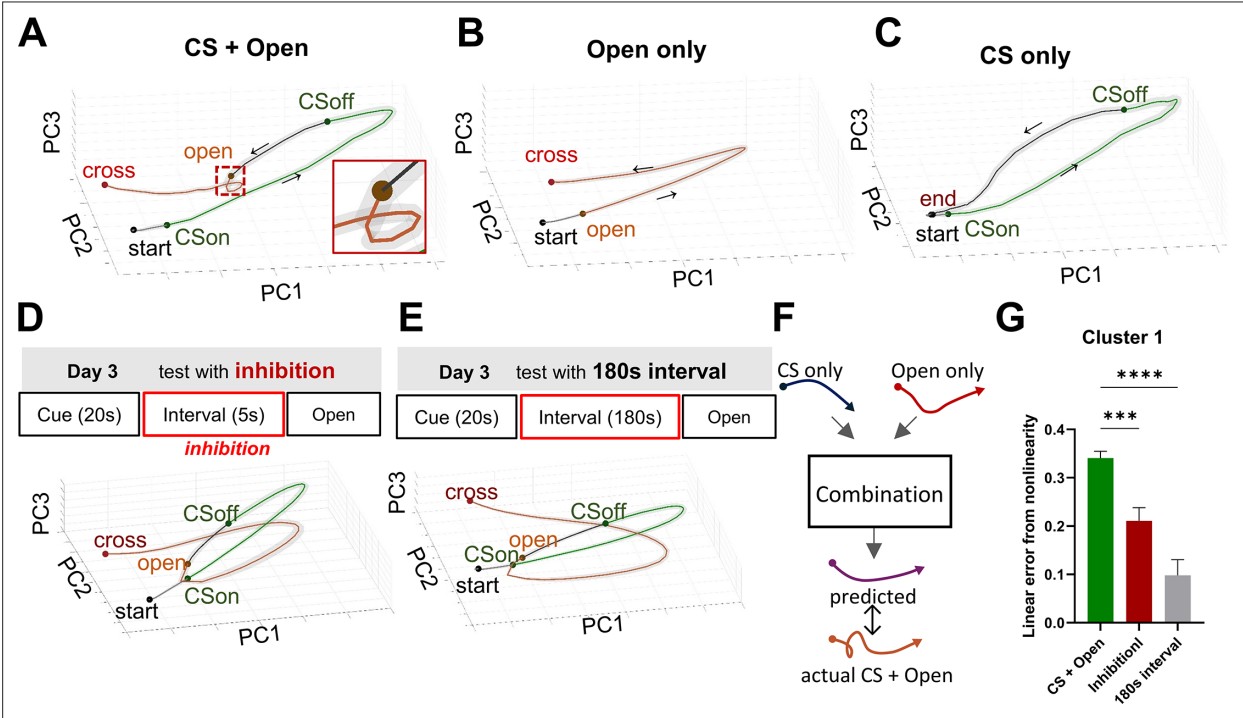

**Figure 3.** RNN PCA trajectories. (**A–C**) Time-normalized, trial-averaged PCA trajectories for each simulation condition. All trajectories begin at the Start event (black circle), proceed through CS onset and offset (CSon and CSoff; green circle) in the CS-containing and inhibition simulations, then through the Open event (orange circle) in the Open-containing and inhibition simulations, and terminate either at the Cross event (red circle) for crossing conditions or at the End event (black circle) for the CS-only condition. (**A**) PCA trajectory for the CS + door-opening condition. The mean latency from door opening to crossing was 44.16 ± 0.9583 s. (**B**) PCA trajectory for the door-opening only condition. (**C**) PCA trajectory for the CS only condition. (**D**) Simulated inhibition applied during a 5-s interval between CSoff and door-opening. Top: schematic of the simulated task. Bottom: time-normalized, trial-averaged PCA trajectory. (**E**) Simulation with a 180-s interval between CSoff and door-opening (no inhibition applied). Top: schematic of the simulated task. Bottom: time-normalized, trial-averaged PCA trajectory. (**F**) Schematic of the trajectory-combination model: the predicted CS + door-opening trajectory (purple line)—obtained by model of which input are the CS only and door-opening only trajectories—is plotted against the actual CS +door-opening trajectory (orange line). (**G**) Model-fit comparison for Cluster 1: difference in residual sum of squares (ΔRSS) between the linear regression and multilayer perceptron (MLP) models, normalized to the mean RSS of the linear model. Bar colors denote condition (CS + door-opening = green, inhibition = red, 180-s interval = gray). One-way ANOVA with Holm–Sidak's multiple comparisons test (CS + door-opening, *n* = 147; inhibition, *n* = 119; 180-s interval, *n* = 128): CS + door-opening vs inhibition, **p = 0.0063; CS + Open vs 180-s interval, ****p < 0.0001. Bars show mean ± SEM. Solid lines represent mean PCA trajectories; shaded areas denote SEM.

The online version of this article includes the following figure supplement(s) for figure 3:

**Figure supplement 1.** PCA trajectories of Clusters 2 and 3 under various simulation conditions.

**Figure supplement 2.** Model-fit comparison across clusters.

In the CS + door-opening condition, the mean trajectory moved away from the start region after CS presentation and progressed along a curved path. Notably, a short-loop pattern appeared immediately after the door opened—5 s after CS offset (*Figure 3A*). In contrast, in the door-opening only group, the trajectory remained near the start region until the door opened and then followed a long-curved path toward the crossing point (*Figure 3B*). As expected, the CS only group exhibited a trajectory similar to that of the CS + door-opening condition until 5 s after CS offset, after which it returned toward the start region (*Figure 3C*). Interestingly, in two manipulations known to delay crossing latency—(*Crick and Koch, 2005*) inhibition of neural activity during the 5-s interval between CS offset and door opening, and (*Shelton et al., 2025*) extension of this interval to 180 s—the short loop observed in the CS + door-opening group was not observed (*Figure 3D and E*).

Sudden changes in trajectory shape, such as the short loop observed in the CS + door-opening condition, have been reported in previous studies of other brain regions to be associated with specific brain functions (*Voigts et al., 2025*). Therefore, based on the assumption that the persisting CS signal and the door-opening signal combine to drive integration, we examined their interaction during this period. To test this, we assumed that the post-CS segment of the CS only trajectory (starting 5 s after CS offset) and the post-door-opening segment of the door-opening only trajectory each represent the respective inputs to the integration process. We then tested whether their combination could explain the post-door-opening trajectory in the CS + door-opening condition (*Figure 3F-G*, *Figure 3—figure supplement 2*). A linear combination model failed to reconstruct the early time bins immediately after door opening, likely due to the distinctive short-loop shape. In contrast, a nonlinear model (multilayer perceptron, MLP) achieved a lower residual sum of squares (RSS), outperforming the linear model (*Figure 3G*, *Figure 3—figure supplement 2A*). The difference in model fit between the nonlinear and linear models was significantly larger in the CS + door-opening condition than in either the inhibition or 180-s interval conditions (*Figure 3G*). Notably, the difference in the CS + door-opening condition was specific to the claustrum-like cluster, as the other two clusters showed no comparable difference (*Figure 3—figure supplement 2B–D*). Taken together, these results suggest that nonlinear integration occurs in the claustrum-like cluster specifically during the short-looping period immediately following door opening in the CS + door-opening condition.

## Trajectory features predicted by the RNN are present in biological recordings

We next asked whether these trajectory features predicted by the RNN are also present in biological claustral recordings. As mentioned earlier, our previous study (*Han et al., 2024*) was limited by the use of single-trial testing, a small number of recorded units per rat, and a limited total number of animals, which restricted both the overall population size and the feasibility of in-depth analyses. To address these limitations, we constructed a trained RNN model that produced reproducible and testable trajectories, particularly those exhibiting short-loop dynamics suggestive of information integration. To compare the model-derived trajectories with empirical neural data, we applied Gaussian Process Factor Analysis (GPFA) (*Yu et al., 2009*; *Lakshmanan et al., 2015*), a dimensionality reduction method well-suited for single-trial time-series data. Although GPFA is typically optimized with multiple trials, it can still provide a useful low-dimensional representation of neural activity. By applying GPFA to single-trial unit data collected from multiple animals and embedding it into a three-dimensional latent space, we were able to visualize population dynamics in a manner comparable to the trajectories generated by the RNN, facilitating comparison between model dynamics and experimental observations (*Figure 4A*).

Interestingly, the trajectory derived from units recorded in CS rats—the experimental group corresponding to the CS + door-opening condition in the RNN model—resembled the short-loop structure observed in the RNN (*Figure 4A*, left). The onset of the short loop was markedly delayed compared to that in the claustrum-like cluster of the RNN. In contrast, trajectories from neutral rats, which received a neutral CS (a light cue not associated with an electrical shock) during testing, exhibited a gentler curvature (*Figure 4A*, right). To confirm the reliability of this structure, we partitioned the neurons into two subgroups and independently repeated the GPFA analysis. Neurons with comparable basal firing rates were paired and subsequently classified into two groups. To compare trajectories before and after the split, the behavioral reference points on the post-split trajectories were aligned to those on the pre-split trajectories. The resulting latent trajectories remained qualitatively similar in shape and

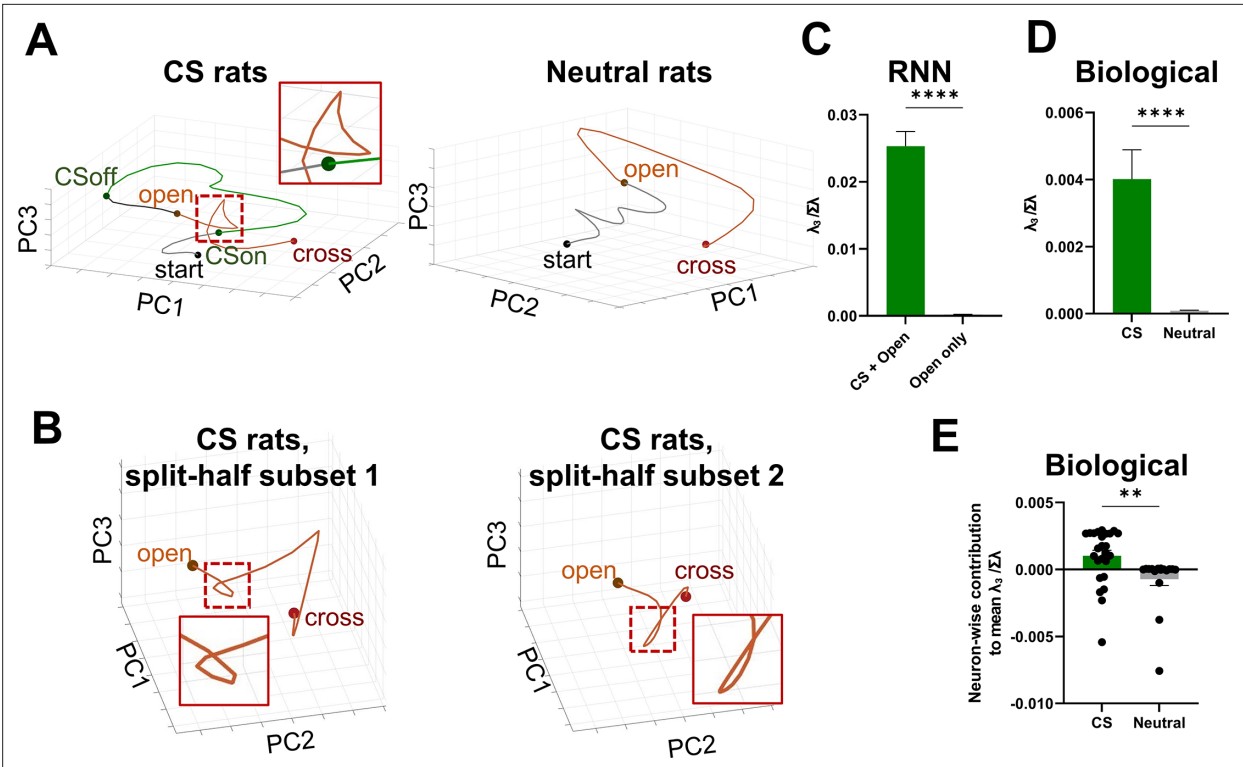

**Figure 4.** Biological neural GPFA trajectories and quantification of local geometry. (**A**) GPFA trajectories of in vivo single-unit recording data: (left) CS rats, (right) neutral rats (*n* = 25 neurons, 16 rats). The mean latency from door opening to crossing was 56.363 ± 8.22 s. (**B**) Validation of GPFA trajectories in CS group recordings. Neurons were ranked in descending order of mean z-scored firing rates (baseline to 5 s post-crossing) and split into odd- and even-indexed subsets. The trajectory from subset 1 (left; *n* = 13 neurons, 6 rats) served as the reference, while the trajectory from subset 2 (right; *n* = 12 neurons, 10 rats) was aligned accordingly. For clarity, only the segment from door opening to crossing is displayed. The mean latency from door opening to crossing for subset 1 was 53.306 ± 14.857 s. The mean latency from door opening to crossing for subset 2 was 50.969 ± 7.417 s. (**C**) Third eigenvalue divided by the sum of all eigenvalues (i.e., the proportion of total three-dimensional variance explained by PC3), computed using sliding windows from the post-open RNN trajectory, comparing the CS + Open (green; *n* = 20 windows) and Open-only (gray; *n* = 20 windows) conditions. **** Unpaired *t*-test, p < 0.0001. (**D**) $\lambda_3/\Sigma\lambda$ of the biological neural trajectory within the sliding window, comparing the CS (green) and Neutral (gray) group pseudopopulations. **** Mann–Whitney test, p < 0.0001. (**E**) Neuron-wise contribution to the mean $\lambda_3/\Sigma\lambda$, defined using a leave-one-neuron-out analysis of the biological neural trajectory, comparing CS (green; *n* = 25 neurons) and Neutral (gray; *n* = 17 neurons). ** Mann–Whitney test, p = 0.0012. Bars show mean ± SEM. Raw data from Figure 4 of ***Han et al., 2024***, *Cell Reports*, were reanalyzed using GPFA. Elsevier.

evolution (***Figure 4B***), suggesting that the short-loop structure reflects coordinated activity across the neuronal population rather than being driven by a small subset of neurons.

To further examine this similarity quantitatively, we performed PCA on the trajectories and quantified their local geometry using an eigenvalue-based metric ($\lambda_3/\Sigma\lambda$) (***Figure 4C–E***). This metric reflects the proportion of variance not explained by the plane defined by PC1 and PC2, thereby capturing deviations from planarity in the trajectory. Importantly, this geometric metric does not rely on recovering neuron-to-neuron noise correlations. Nevertheless, the estimation of the latent space still depends on the covariance structure among claustral neurons, suggesting that the inferred trajectories remain tied to biologically meaningful population dynamics. When applied to the PCA trajectories of the RNN, the metric was significantly higher in the CS + Open condition than in the Open-only condition (***Figure 4C***). Similarly, when the GPFA trajectories from real claustral neurons were reanalyzed using PCA and the same metric was applied, the CS group showed significantly higher values than the Neutral group (***Figure 4D***). Furthermore, a leave-one-neuron-out analysis revealed that claustral neurons in the CS group contributed more uniformly to this metric compared to those in the Neutral group (***Figure 4E***). These findings support the interpretation that the observed short-loop structure did not arise from a small subset of neurons, outliers, or chance fluctuations.

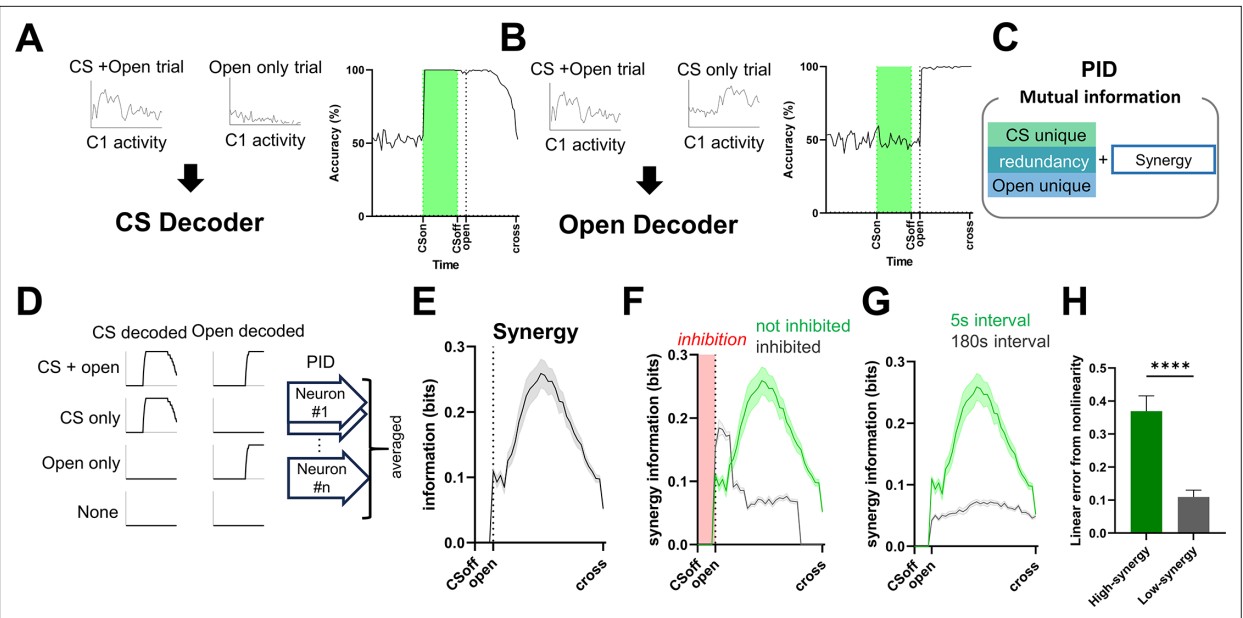

**Figure 5.** Decoder accuracy, and Partial Information Decomposition of Cluster 1 RNN. Neurons under different simulation conditions. (**A**) Left: schematic of the CS decoder. A classifier was trained to distinguish CS + door-opening trials from door-opening only trials using Cluster 1 neuron activity. Right: CS decoding accuracy (%) across time bins. (**B**) Left: schematic of the door-opening decoder. A classifier was trained to distinguish CS + door-opening trials from CS only trials using Cluster 1 neuron activity. Right: door-opening decoding accuracy (%) across time bins. (**C**) Conceptual diagram of Partial Information Decomposition. Mutual information about CS and door-opening is decomposed into CS-unique information, door-opening-unique information, redundancy, and synergy. (**D**) Schematic of the PID analysis. CS decoding accuracy and door-opening decoding accuracy were computed for each trial type (CS + door-opening, CS only, door-opening only, and None), and used as input variables for neuron-wise PID. PID terms were then averaged across neurons. (**E**) Synergy information of Cluster 1 neurons. (**F**) Comparison of synergy between inhibition and no-inhibition simulations. (**G**) Comparison of synergy between the 180-s interval and the 5-s interval simulations. Green line: 5-s interval; green shaded box indicates CS presentation window. Gray line: 180-s interval. Solid lines represent means; shaded areas indicate SEM. (**H**) Model fit comparison between Cluster 1 RNN neurons with high synergy and those with low synergy. The difference in residual sum of squares (ΔRSS) between linear regression and multilayer perceptron (MLP) models is shown, normalized to the mean RSS of the linear model. Bar colors indicate condition (high synergy = green, low synergy = gray). (High synergy: $n$ = 147 trials; Low synergy: $n$ = 147.) Mann–Whitney test: high vs low synergy, ****$p < 0.0001$. Bars show mean ± SEM. In this figure, Open denotes door-opening.

The online version of this article includes the following figure supplement(s) for figure 5:

**Figure supplement 1.** Decoding performance and PID analysis across clusters.

**Figure supplement 2.** PCA trajectories of high- and low-synergy neurons.

## Partial information decomposition uncovers neurons carrying synergy between CS and door-opening signals

Having established that nonlinear integration emerges at the population level, we next asked how this integration is implemented at the level of individual neurons. Because this analysis requires full population observability and repeated trials, which are not feasible in the biological dataset, we performed the following analyses using the trained RNN model. To examine the contribution of individual neurons to this nonlinear integration, we measured the synergistic information for each neuron—i.e., information encoded only when both stimuli, CS and door opening, were present. To this end, we performed Partial Information Decomposition (PID) analysis (*Timme and Lapish, 2018*; *Timme et al., 2014*). PID quantifies how two input variables (CS and door opening) contribute to the information contained in the output neural activity by decomposing the total mutual information into three components: (*Crick and Koch, 2005*) unique information provided independently by each input, (*Shelton et al., 2025*) redundant information shared by both inputs, and (*White et al., 2018*) synergistic information generated only when both inputs are present simultaneously (*Figure 5C*).

In the trained RNN model (as well as in real claustral neurons), the effect of an input stimulus persists in neural activity even after the physical stimulus has ended. Therefore, to define stimulus strength for PID analysis, we used the output of trained decoders—which infer the presence or absence of each

stimulus from neural activity—as the effective input variable in the PID computation. We trained two decoders using the z-scored activity of neurons in the RNN claustrum-like cluster: one to distinguish between CS-present and CS-absent trials, and another to classify door-open vs no-door-open trials (*Figure 5A, B*, *Figure 5—figure supplement 1*; see Methods). The CS decoder performed at chance level during the baseline period but sharply increased in accuracy immediately after CS onset, with performance remaining high well beyond the end of CS presentation (*Figure 5A*). The door-opening decoder showed high accuracy during the period when the door signal was present (*Figure 5B*).

We used the CS decoding accuracy score and door-opening decoding accuracy score as inputs, and the *Z*-scored activity of claustrum-like cluster neurons in each trial as the output. PID was computed for each neuron over time (*Figure 5D–H*, see Methods). As expected, CS-unique information rose following CS onset and gradually decayed, whereas door-opening-unique information increased after door opening (*Figure 5—figure supplement 1*). Importantly, synergistic information—representing new information generated only when CS and door-opening representations were jointly present—increased immediately after door opening and then gradually decreased (*Figure 5E*). This suggests that neurons in the claustrum-like cluster integrate CS and door-opening signals to produce new, combined information. Moreover, synergistic signals were reduced in two conditions where nonlinear integration was disrupted: when inhibition was applied to the network, and when the CS–door interval was extended to 180 s (*Figure 5F, G*).

We further divided claustrum-like cluster neurons into the top 25% with the highest synergy ('high-synergy' neurons) and the bottom 25% ('low-synergy' neurons) and analyzed their trajectory features (i.e., the short loop). In high-synergy neurons, the curvature of the neural trajectory immediately after door opening exhibited a more pronounced turning-angle change compared to low-synergy neurons (*Figure 5—figure supplement 2A, B*). Furthermore, when CS + door-opening trials were predicted using trajectories from CS-only and door-opening-only conditions, nonlinear prediction errors were greater for high-synergy neurons than for low-synergy neurons (*Figure 5H*). Together, these findings suggest that synergistic information at the single-neuron level is related to the short loop and that, within the claustrum-like cluster, certain neurons play a more dominant role in the nonlinear integration process.

## Trajectory-based dynamic coding underlies integration

We next used the RNN model to examine how this integrated representation is dynamically encoded over time. To assess the temporal evolution and stability of the integrated representation, we applied cross-temporal decoding. This approach allowed us to examine whether the combined information from CS and door-opening is encoded in a static manner or in a time-varying fashion.

Using cross-temporal decoding, we examined how the neuronal population encodes information unique to the CS + door-opening condition over time. For each corresponding time bin across the three conditions (CS + Door, CS-only, and Door-only), we trained a decoder to distinguish the CS + Door condition from the other two, and evaluated its classification accuracy across all time bins. In the entire population of claustrum-like cluster, cross-temporal decoding revealed a mixed coding regime: shortly after door opening, decoders trained at specific time bins maintained discrimination power across relatively broad temporal windows (*Figure 6A*). However, as time progressed, the discriminative power of decoders became temporally confined, performing well only near the training bin.

When neurons in the claustrum-like cluster were divided into high- and low-synergy groups and decoders were trained and tested separately on each, high-synergy neurons exhibited a dynamic coding pattern. Decoders trained on specific time bins in this group showed strong performance at that time point, with limited generalization to neighboring bins. In contrast, decoders trained on low-synergy neurons maintained more temporally stable performance. These results suggest that high synergy neurons, which contribute critically to nonlinear integration, encode CS and door-opening information using a dynamic coding scheme, whereas low-synergy neurons may support a more stable representation (*Figure 6B–C*).

This distinction was further supported by raw firing-rate heatmaps: both high and low synergy neurons showed increased activity following CS onset, but after the door opened, only high-synergy neurons displayed variable firing patterns (*Figure 6D*, left). Notably, a similar pattern of transient, variable spiking was observed in our in vivo recordings of claustral single units after door opening in the CS + door-opening group (*Figure 6D*, right).

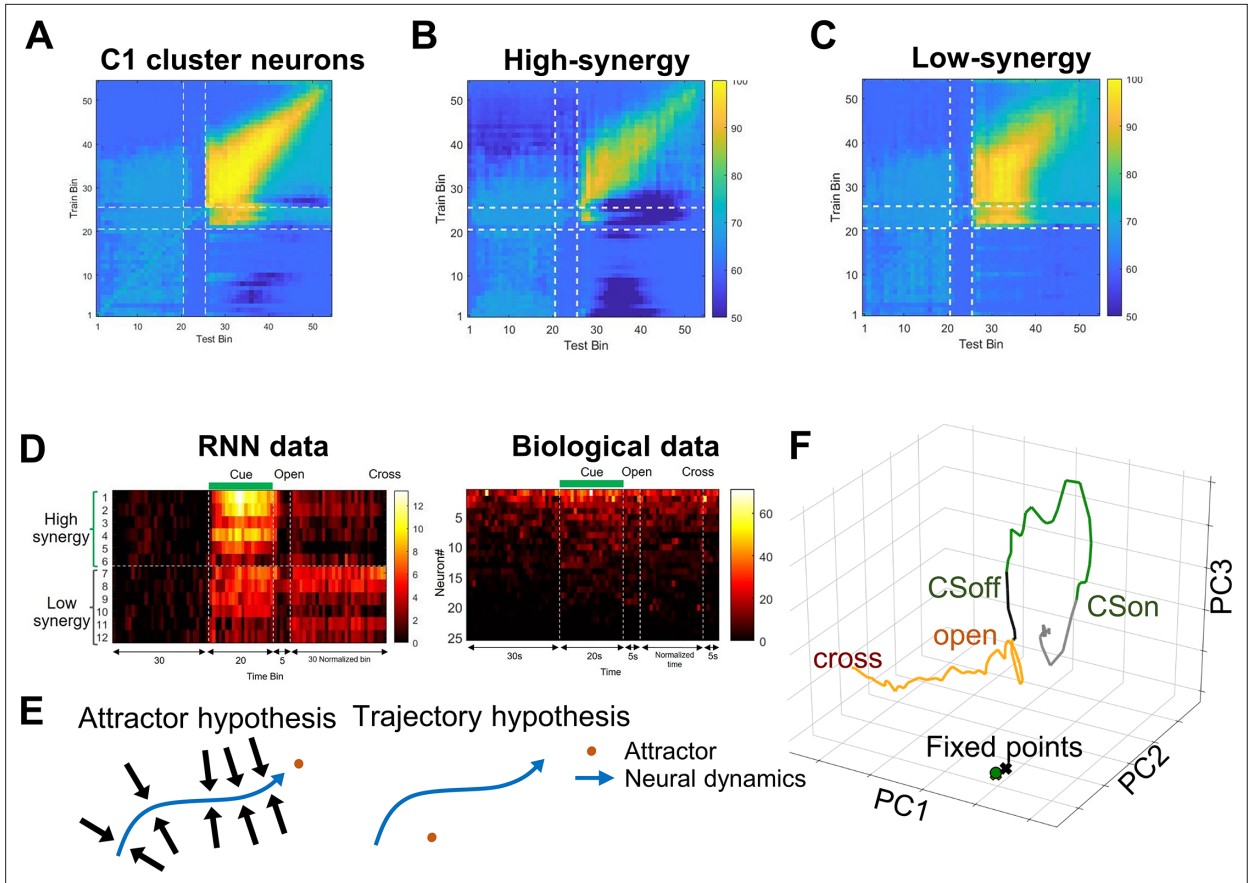

**Figure 6.** Trajectory coding hypothesis and biological neural trajectories. (**A**) Cross-temporal decoding of integration-specific information from Cluster 1 RNN neurons. (**B**) Cross-temporal decoding of integration-specific information from Cluster 1 RNN neurons with high synergy. (**C**) Cross-temporal decoding of integration-specific information from Cluster 1 RNN neurons with low synergy. (**D**) Left: heatmaps of raw firing rates for RNN neurons from a representative CS + door-opening trial. Right: heatmaps for biological Cluster 1 non-exploratory neurons in the CS group during the delayed escape task (right). Neurons in the right heatmap are ordered by overall activity. (**E**) Hypotheses on dynamic coding. Left: attractor hypothesis: multiple or continuous stable converging points exist, and changes in input cause neural dynamics to move from one point to another, thereby encoding varying states. Right: trajectory hypothesis: neural dynamics are not, or are only minimally, related to attractors (which may not exist); instead, changes in input cause changes in trajectory, which encode varying states. (**F**) Fixed-point analysis of a representative trial in an RNN. Green dots represent stable fixed points (CS period), while black and yellow X-marks denote unstable fixed points (interval and open periods). Panel D (right) is adapted from Fig. S4E of *Han et al., 2024*, *Cell Reports*, Elsevier.

Temporal evolution of the population state can be framed by two hypotheses (*Figure 6E*). The attractor hypothesis posits that neural dynamics consist of discrete or continuous attractors, with the integrated state represented by transitions among them in response to varying inputs. The trajectory hypothesis proposes that integration is embodied in the evolving trajectory itself, independent of fixed attractors. To distinguish these views, we performed fixed-point analysis on the RNN, locating states where the velocity vector is nearly zero (*Golub and Sussillo, 2018*). Fixed points were found to lie far from the empirical trajectories, indicating that the network's dynamics are trajectory-based rather than attractor-based when integrating CS and door-opening cues (*Figure 6F*).

## Discussion

In this study, we trained a vanilla RNN solely on behavioral metrics from the delayed escape task, which had previously been shown to require rsCla (*Han et al., 2024*). The trained network developed a claustrum-like cluster whose dynamics showed similarities to the experimentally observed population activity patterns and reflected the relationship between neural activity and behavior. Low-dimensional trajectory analysis revealed a nonlinear integration of the CS and the temporally delayed door-opening signal, accompanied by an increase in synergistic information. Rather than converging to

stable attractor states, the network encoded these signals through continuously evolving trajectories. In the post-integration phase, the trajectories formed a sharply curved 'short loop' immediately after door opening—a trajectory pattern also observed in real claustrum recordings, thereby supporting the biological plausibility of the model. Such temporal integration through dynamic coding enables flexible responses to inputs that arrive in the claustrum with temporal gaps and offers the advantage of generating richer representations compared with stable coding (*Stroud et al., 2023*; *Brody et al., 2003b*; *Chaisangmongkon et al., 2017*; *Stroud et al., 2024*; *Voigts et al., 2025*; *Brody et al., 2003a*). Moreover, contrary to the intuitive expectation that dynamically coded information might be difficult to decode, it can, in fact, be easily read out using simple linear decoders (*Meyers, 2018*), which provides an additional advantage.

It is important to emphasize that behavior-trained RNNs can admit multiple internal solutions capable of producing the same behavioral output. Accordingly, the network analyzed in this study represents only one possible computational realization. Nevertheless, the dynamical regime observed in the present model converged with several independent lines of evidence from claustral recordings. These include persistent neural activity during the delay period, the correlation between neural activity and escape latency, recurrent connectivity, and the consistent geometric changes in population trajectories revealed through population-level analyses. Taken together, these findings suggest that the present computational model should not be interpreted as a direct implementation of the claustrum's computational mechanism, but rather as a candidate model that captures a plausible computational principle that may operate in the claustrum. In particular, conclusions derived from the RNN analysis, such as increased synergy and dynamic coding, remain hypotheses that could not be directly validated in the biological claustrum due to experimental constraints, and therefore require future experimental testing. At the same time, the present model was specifically designed to capture the computations required for the delayed escape task and does not aim to account for the full range of claustral functions. In particular, it is not intended to replace or encompass the diverse roles of the claustrum proposed in previous studies, including those from the Mathur and Citri research groups (*White et al., 2018*; *Atlan et al., 2024*; *Faig et al., 2024*; *Niu et al., 2022*; *Chevée et al., 2022*; *Narikiyo et al., 2020*; *White et al., 2020*; *Qadir et al., 2022*; *Goll et al., 2015*; *Atlan et al., 2018*; *Terem et al., 2020*; *Madden et al., 2025*). Future studies will be needed to determine whether dynamical principles similar to those identified in the present model also contribute to other proposed functions of the claustrum, including attention, salience detection, cognitive control, sleep, and premotor control.

Previous studies have shown that the anterior claustrum is largely unresponsive to simple sensory stimuli, becoming active only when such stimuli acquire behavioral relevance possibly through convergent inputs from regions encoding memory, value, context, and motivational states (*Chevée et al., 2022*; *Han et al., 2024*; *Park et al., 2025*; *Naghavi et al., 2007*; *Reus-García et al., 2021*; *Ollerenshaw et al., 2021*; *Stewart et al., 2024*). It also exhibits stronger responses to cross-modal stimuli when they are semantically congruent (*Naghavi et al., 2007*; *Froesel et al., 2024*). These findings suggest that the claustrum preferentially integrates behaviorally meaningful inputs via nonlinear operations to generate unified representations. Within this framework, both the CS and the door-opening cue in the delayed escape task may carry escape-relevant information. However, since the behavioral meaning of the door-opening cue may require additional processing in the biological circuit, the integration process may require additional time, which could explain the delayed emergence of the short-loop trajectory in real neural data (see *Figure 4*, left). This interpretation may be viewed as broadly compatible with prior theoretical proposals suggesting that the claustrum contributes to the integration of semantically related information (*Crick and Koch, 2005*). Furthermore, such integrated information is likely broadcast to higher-order cognitive regions such as the PFC, ACC, and OFC, where it can be read out and interpreted (*Seguin et al., 2023*), a possibility that resonates with broader theoretical accounts of large-scale integrative brain function (*Crick and Koch, 2005*; *Dehaene et al., 1998*).

Although our conclusions are mainly derived from modeling, the correspondence between model predictions and empirical observations provides a rationale for future experimental validation. Multi-trial behavioral paradigms and large-scale single-neuron recordings will be essential to rigorously test these predictions. Furthermore, elucidating the nature of the preprocessing that shapes inputs to the claustrum, as well as the downstream pathways through which its integrated

representations are accessed and utilized, will be critical for understanding its role in brain-wide computation.

## Materials and methods

### Animals

The following behavioral procedures were previously reported in *Han et al., 2024* and are summarized here for clarity (*Han et al., 2024*). Male Long-Evans rats (Japan SLC Inc) were dual-housed for 5–11 days before the experimental procedures began, under 12 hr inverted light/dark cycle (light off at 9:00 a.m.) with ad libitum access to food and water. All behavioral experiments were conducted during the dark portion of the cycle. The Institutional Animal Care and Use Committee (IACUC) of the Seoul National University approved all experimental procedures (SNU-160712-6-1), and all experiments were performed following the guidelines for care and use of laboratory animals of the Seoul National University. The in vivo single-unit recording experiments for each of the CS and Neutral groups were performed separately.

### Delayed escape task with in vivo single-unit recording

The following recording procedures were previously reported in *Han et al., 2024* (https://doi.org/10.1016/j.celrep.2024.113748) and are summarized here for clarity (*Han et al., 2024*). Rats (7 weeks old) were anesthetized with an intraperitoneal (i.p.) injection of sodium pentobarbital (50 mg/kg) and maintained with isoflurane (1–1.5%) in $O_2$. Rats were mounted on a stereotaxic apparatus (Stoelting Co). Fixed-wire electrodes were bilaterally implanted into the rsCla (AP +3.35 /ML ±2.10/DV-4.50). A ground wire was implanted in the cerebellum. The electrodes consisted of eight individually insulated nichrome microwires (50 μm outer diameter, impedance 1–3 MΩ; California Fine Wire) contained in a stainless steel guide cannula. The electrodes were affixed to the skull with screws using Poly-F zinc polycarboxylate cement (Konstanz), vertex self-curing (vertex-dental, Zeist), and bond (Loctite 411, Henkel). Rats were allowed to recover for a week before they underwent experimental procedures.

The rats in both the CS and Neutral groups were presented with the green light cue (10 s × 5 times) in the fear conditioning context with modification (a foamex floor instead of a metal grid floor), which allowed for adaptation to the recording cable attachment. After 24 hr, the rats were presented with the green light cue (10 s × 5 times, average inter-trial interval: 90 s) in the same context, to optimize recording parameters for each channel. Approximately 1 hr later, the rats in the CS group were fear-conditioned, and the rats in the Neutral group underwent pseudo-conditioning in which the green light cues were presented but electrical foot shocks were omitted.

On day 2, rats underwent the two-compartment chamber (54 × 23 cm area in total) exposure session. The two compartments had identical floor sizes, but they were differently constructed (compartment A vs B: foamex floor vs mesh grill floor; no visual cued walls vs visual cued walls with circles; 50 vs 30 cm height; a green LED bulb in the middle of the side wall and a speaker right above the LED vs no accessories installed on the wall, respectively). The two compartments were connected by opposing square-shaped (7 × 7 cm) outlets, located 12 cm above the floor. The outlet could be opened or closed by sliding. To ensure that the subject localized a source of light within compartment A, the walls of the compartment were made of non-reflective material (black foamex). Consequently, compartments A and B were illuminated to a lesser intensity than in the fear conditioning performed on day 1, in which the walls were made of reflective materials.

The rats were placed in compartment A to acclimate for 2 min on average on day 2. The outlet in the wall (without any curtains) was then opened simultaneously with a tone pip sound (2.8 kHz, 200 ms, 85 dB). The rats were allowed to cross over to compartment B for a duration of 5 min. Once they crossed, the outlet was closed and the rats were allowed to stay in compartment B for 2 min. The rats were then returned to their home cages. If a rat did not cross in 5 min, it was removed from compartment A and temporarily moved to a small box near the two-compartment chamber. Around 30 s later, the rat was re-introduced to compartment A and the same procedure was performed. The rats that did not cross after 10 trials were excluded from further experiments. We measured crossing latency and the number of trials in which the rat succeeded in crossing. Crossing latency was defined as the duration from outlet opening to the moment when all four paws of the rat touched the floor

of compartment B. The chamber was cleaned with 70% ethanol and then with distilled water prior to the beginning of each trial.

For the delayed escape test session (day 3), the two-compartment chamber was modified to further prohibit any potential association between the outlet opening and crossing behavior. The opened outlet was visually blocked by a black curtain installed at the outlet side of compartment B. The rat head could go through the curtain to see compartment B since the curtain consisted of two pieces of fabric adjoined at the centerline of the outlet. The rats repetitively put their heads into compartment B through the curtain during the test session, but this behavior did not appear to be associated with outlet crossing. In the modified chamber, a 20-s CS was presented 220 s after the placement of the rat in compartment A. The CS presentation was delayed when the rat showed immobility at the time of CS onset. The outlet was opened with a tone pip sound (2.8 kHz, 200 ms, 85 dB) 5 s after the CS offset, and the rat was allowed to cross the outlet for a duration of 5 min. The test session was performed only once. The rats that did not cross during the test session were considered to omit the task, and they were excluded from data analysis.

## RNN modeling

The RNN model was implemented in PsychRNN (**Ehrlich et al., 2021**; **Ehrlich and Murray, 2022**) as a continuous-time network with a 1-s integration step. Each network consisted of a single hidden layer of 100 ReLU units, of which 80% were excitatory and 20% inhibitory.

Hidden-state $h(t)$ dynamics followed

$$\tau \dot{\boldsymbol{h}}(t) = -h(t) + f\left[\boldsymbol{W}_{\text{rec}}\boldsymbol{h}(t) + \boldsymbol{W}_{in}x(t) + \boldsymbol{b}_{\text{rec}}\right] + \boldsymbol{\eta}(t)$$

$$f(x) = \max(x, 0)$$

$$\boldsymbol{y}(t) = W_{\text{out}}\boldsymbol{h}(t) + b_{\text{out}}$$

An internal time constant of $\tau = 1000$ ms was adopted, in accordance with electrophysiological measurements (**Figure 2—figure supplement 1A**), and ReLU activation $f(x) = \max(0, x)$ was used throughout the network. Here, $W_{\text{rec}}$, $W_{\text{in}}$, and $W_{\text{out}}$ denote the recurrent, input, and output weight matrices, respectively, while $b_{\text{rec}}$ and $b_{\text{out}}$ are constant biases applied to the recurrent and output units. Gaussian noise $\eta(t)$ with variance $\sigma^2 = 0.05^2$ was injected into the recurrent layer. Simulations were conducted using a continuous-time RNN with a discretized time step of 1000 ms. To ensure stable signal propagation and gradient flow during early training, the recurrent weight matrix ($W_{\text{rec}}$) and output weight matrix ($W_{\text{out}}$) were initialized using a Glorot normal distribution. Specifically, initial weights were drawn from a normal distribution centered on 0 with a standard deviation of $\sigma = \sqrt{2/(N_{in} + N_{out})}$ where $N_{in}$ and $N_{out}$ denote the number of input and output units for the given connectivity matrix. Input weights ($W_{\text{in}}$) were initially drawn uniformly from the interval [0, 0.15], after which all rows corresponding to inhibitory units were zeroed out to enforce Dale's principle. To strictly enforce Dale's principle throughout training, rows in ($W_{\text{out}}$) corresponding to inhibitory units were zeroed out, and recurrent weights were constrained such that excitatory and inhibitory neurons maintained their respective positive and negative synaptic projections.

We trained the RNN in a supervised manner using the behavioral data from rat experiments (**Han et al., 2024**). We provided the RNN with sequential inputs that matched the experimental timeline: a 30-s baseline period with no input, followed by a 20-s CS input period, then a 5-s interval with no input, and finally a sustained door-open signal until the end of the trial. In the control group (neutral group), no input was provided during the period corresponding to CS presentation, and trials from the CS and neutral groups were presented in a randomized order. The network's output was treated as the probability of crossing; when this probability exceeded 0.5, the model was considered to have initiated a cross, allowing us to compute latency for each trial.

The network received four input channels, denoted as $x(t) = \left[x_{\text{CS}}, x_{\text{door}}, x_{\text{inh}}, x_{\text{ctx}}\right]^{\top}$. To reflect the intense and salient nature of the fear-conditioning experience, the magnitude of the CS channel $(x_{\text{CS}}(t))$ was set to 2 during the presentation of the CS, specifically from $t = 70$ s to $t = 90$ s (20 s duration), serving as a relatively stronger sensory drive compared to the simple mechanical cue of the door opening. The door signal $(x_{\text{door}}(t))$ was set to 1 starting at $t = 90 + d$ s, where $d$ was selected from [0, 1, 2.5, 5] s depending on curriculum learning, and remained 0 beforehand. The inhibition signal $(x_{\text{inh}}(t))$

was set to –1 for the 5 s immediately following the CS offset ($t$ = 90–95 s) on inhibition trials. Finally, the context input ($x_{\text{ctx}}(t)$) was maintained at 1 throughout the entire trial.

Input weights were first drawn uniformly from the interval $[0, 0.15]$, after which all rows corresponding to inhibitory units were zeroed out to enforce Dale's principle.

$$\widetilde{y}(t) = \begin{cases} 0, & t < t_{tar}, \\ 1, & t \geq t_{tar}. \end{cases}$$

A cross was defined as the first time point $t^*$ at which the network output $y(t^*) \geq 0.5$. The desired output trajectory $\widetilde{y}(t)$ was specified as a step function transitioning at the target escape time $t_{\text{tar}}$, which combines the door-opening time and the experimentally measured behavioral latency measured in rats (latency for CS present: 48.7 s, latency for CS absent: 111.3 s):

$$t_{\text{tar}} = \begin{cases} t_{\text{door}} + \ell_{\text{CS}} = 90 + d + 48.7 \text{ s}, & \text{CS present,} \\ t_{\text{door}} + \ell_{\text{noCS}} = 90 + d + 111.3 \text{ s}, & \text{CS absent.} \end{cases}$$

Using fixed empirical latencies anchored the network to the average rat behavior, such that remaining variability in output timing arose solely from the network's internal dynamics.

The per-trial loss function minimized during training was:

$$L = -\sum_t \left[ \widetilde{y} \ln y + \left(1 - \widetilde{y}\right) \ln\left(1 - y\right) \right] + \lambda \left( \|\boldsymbol{W}_{\text{in}}\| + \|\boldsymbol{W}_{rec}\| \right) + \lambda_{FR} \left\langle \|\mathbf{h}(t)\|_2^2 \right\rangle_t$$

The L2 weight regularization coefficient $\lambda = 0.01$ and the firing rate regularization coefficient $\lambda_{FR} = 0.95$. were systematically optimized via a grid search procedure to maintain biologically plausible firing rates while preventing overfitting. The network was trained using TensorFlow 2.1.0 and the Adam optimizer (learning rate $\alpha = 5 \times 10^{-4}$) for 1.2 million iterations with a batch size of 256. Each training batch comprised trials assigned randomly to CS-present or CS-absent conditions with equal probability (50% each). The delay d between CS offset and door-opening was incrementally increased (0, 1, 2.5, and 5 s) once the batch-wise accuracy exceeded thresholds of 0.5, 0.5, 0.5, and 1, respectively. All simulations were executed on a single RTX-2080 Ti GPU using XLA just-in-time compilation.

After training was completed, the network was saved for downstream testing under various experiment conditions, including inhibition, extended (180 s) interval, CS-only, no-CS/no-Door-opening, and stimulation of the slice experiments. The resulting cross latencies $t^*$, neuronal activity $\mathbf{h}(t)$, and weight matrices were exported to MATLAB for further analyses such as clustering, PCA, and PID.

## RNN population analysis

The RNN simulation was processed with the same population-level pipeline used for the experimental data (*Han et al., 2024*). The first 10 s of each simulation were discarded to eliminate variability caused by initial states. Activity from the next 30 s served as the baseline for Z-score normalization.

For every CS trial ($n$ = 147), Z-scores were averaged neuron-wise and then summarized across five task epochs—cue, interval, early-open, late-open, and the 5 s after the cross. These epoch-wise vectors were embedded into a three-dimensional t-SNE space (perplexity = 24, exaggeration = 48, RNG = mt19937 ar). The optimal number of clusters was determined with the gap statistic and applied in a k-means clustering step. Through this procedure, we obtained three clusters consisting of 24, 38, and 38 units (with 24 units in the claustrum-like cluster). Here, the term claustrum-like refers to the fact that the mean response of the units remained persistent not only during the CS presentation but also throughout the subsequent period until crossing. The main difference between the empirical experiment and the RNN results is that, in the actual experiment, each rat was tested with only one latency value, so correlations had to be calculated across animals, whereas in the simulation, multiple trials were generated from a single RNN, allowing correlations to be calculated across trials. Specifically, for each trial, the mean Z-scored activity across the 24 units during the cue-to-cross window was calculated, and this value was paired with the crossing latency from that trial., and these Z-scores were paired with the crossing latency from that trial. With 100 trials, this yielded 100 data points. The

variability of model latency reflects both the injected noise and differences in initial states. Notably, only a subset of the supervised RNNs we trained produced a claustrum-like cluster; among 100 independent runs, only five met this criterion, and all results reported here are based on one representative RNN network from those five cases.

As shown in *Figure 1—figure supplement 2*, the output contribution of each cluster for each time points was computed as the product of the hidden-state vector $\mathbf{h}(t)$ and the output-weight matrix $W_{out}$.

## Slice electrophysiology

Male Long-Evans rats (3–4 weeks old) were anesthetized (sodium pentobarbital, 50 mg/kg, i.p.) and mounted in a stereotaxic frame. For optogenetic circuit mapping, AAV encoding ChrimsonR (pAAV-CamKIIa-ChrimsonR-mScarlet-KV2.1; Addgene; $1 \times 10^{13}$–$1 \times 10^{14}$ vg/ml) was diluted 1:4 in sterile saline for sparse labeling and injected unilaterally (200 nl) into the rsCla (AP +2.95, ML ±1.95, DV −3.85 mm). For calcium imaging, AAV encoding GCaMP6f (pENN.AAV.CamKII.GCaMP6f.WPRE.SV40; Addgene; ≥$1 \times 10^{13}$ molecules/ml) was injected bilaterally (400 nl) into the same coordinates. Viruses were delivered at 20 nl/min via a pulled glass capillary using a pressure injection system (Nanoject II, Drummond; or Nanoliter 2010, WPI). The pipette was left in place for 10 min post-infusion to ensure diffusion. Animals were dual-housed for 3–4 weeks for viral expression. Subsequently, rats were anesthetized with isoflurane and decapitated for acute slice preparation. The isolated whole brains were placed in the slicing chamber filled with a warm (34–36°C) artificial cerebrospinal fluid (aCSF) solution containing 120 mM NaCl, 3.5 mM KCl, 1.25 mM $NaH_2PO_4$, 26 mM $NaHCO_3$, 1.3 mM $MgCl_2$, 2 mM $CaCl_2$, and 11 mM D-(+)-glucose, and continuously bubbled at room temperature with 95% $O_2$/5% $CO_2$. Acute sagittal or horizontal slices containing the rsCla (300 µm thickness) were prepared using a vibratome (VT1200, Leica). Slices were then stored under submerged conditions at 34°C for 1.5 hr before recordings. For recording, slices were transferred to the recording chamber in which aCSF was continuously perfused. The perfused aCSF was continuously aerated with 95% $O_2$/5% CO, and maintained at 32°C.

Whole-cell patch clamp recordings were performed using micropipettes (4–6 MΩ) which were pulled from borosilicate glass capillaries (1.2 mm OD, 0.94 mm ID, Warner Instruments; Massachusette, USA) with a micropipette puller (Pc-10, Narishige). The pipettes were filled with the following solution: 120 mM potassium D-gluconate, 0.2 mM EGTA, 10 mM HEPES, 5 mM NaCl, 2 mM Mg-ATP, 0.3 mM Na-GTP, and 1 mM $MgCl_2$, with the pH adjusted to 7.2 with KOH and osmolality adjusted to approximately 297 mmol/kg with sucrose. Recordings were made under infrared differential interference contrast–enhanced visual guidance from neurons located in the rsCla slices. Neurons were first current-clamped neurons and then used with membrane potentials lower than –50 mV. The neurons were voltage-clamped at –70 mV, and solutions were delivered to slices via superfusion driven by gravity at a flow rate of 1.3 ml/min. The pipette series resistance was monitored throughout the experiments, and if it changed by >20%, the data were discarded. To avoid direct light stimulation, whole cell recordings were conducted using neurons in which the expression of ChrimsonR was absent. Wide-field photostimulation was executed via a microscope's objective lens (40×/0.80NA) using a power of 6.4 mW (measured at the lens level) and a wavelength of 617 nm. This stimulation was achieved through a digital mirror device (Polygon 400, mightex) for a duration of 2 ms. To stimulate each of the divided parts of the entire optical field, the same digital mirror device was used. The entire optical field was partitioned into a 30 by 30 grid, yielding a total of 900 equivalent sections. Each section received 2 ms of stimulation, and the heatmap was generated using the Matlab interpolation algorithm with the EPSC data. The resultant EPSC amplitudes were used to generate an EPSC heatmap encompassing the entire field. Whole-cell currents were filtered at 2 kHz, digitized at up to 10 kHz, and stored on a microcomputer (Clampex 10.7 software, Molecular Devices). We used 10 µM of NBQX (Tocris), and 50 µM of D-AP5 (Tocris). One or two neurons were recorded per animal (a single neuron per slice). All recordings were completed within 5 hr after slice preparation.

In brain slices where GCaMP6f was expressed, electrical stimulation was applied using a Concentric Bipolar Electrode (FHC) via an isolator (WPI, A360). Stimulation was delivered at 150–200 µA, 20 Hz, 1-s duration. The puffer pipette was positioned 75 µm below the slice surface and 50 µm away from the electrode. At the 10-s mark post electrical stimulation, pressure injection was carried out under a pressure of 20 psi. We used 100 µM of NBQX (Sigma) and 500 µM of D-AP5 (Tocris) for this

experiment. The analysis of fluorescence changes in the recorded images follows these steps: Initially, cell boundaries were manually marked for the 35 recorded GCaMP6f slice samples. Subsequently, a U-Net deep learning model was trained on these images to develop a cell segmentation model. Using this model, the attached cells were segmented and separated utilizing the watershed algorithm. Finally, all cells were manually reviewed once more through video inspection.

## RNN simulation adjusted for slice-stimulation conditions

To mimic the effect of electrical stimulation and applied NBQX + D-AP5 in the acute slice, we constructed a reduced circuit composed exclusively of Cluster 1 (C1) units. All recurrent weights outside C1 were zeroed, leaving only C1 → C1 connections in $W_{rec}$. Because inhibitory neurons may lose efficacy in acute slices, every inhibitory weight in $W_{rec}$ was scaled to 40% of its original value.

Stimulation was modeled by injecting an external drive $x_{\text{stim}}(t) = 2$ for 1 s. The corresponding input weights were drawn uniformly from 0 to 0.15 for excitatory neurons only. Input weights to the inhibitory neurons were set to zero.

In our model, we simulated the effects of applying NBQX + D-AP5 by introducing recurrent inhibition, which began 10 s after the initial stimulation. The inhibition was targeted, affecting only 10% of the neurons in the network. For these selected neurons, the strength of the excitatory inputs they received from other neurons ($W_{rec}$) was reduced by 60%. This simulation was designed to replicate key findings from our slice experiments (*Figure 2—figure supplement 1A*): First, the effect of NBQX + D-AP5 is localized and does not spread widely; and second, drug application resulted in an approximately 60% reduction in the amplitude of EPSCs.

## Visualization of PCA trajectories

To visualize neural dynamics, trial-wise Z-scored firing rates from a specific neuronal cluster were extracted for each simulation group (CS + Open, Open-only, and CS-only). Trials were concatenated across groups and reshaped into a 2D matrix of size (trial x time) × neuron for PCA. PCA was performed on the combined matrix, and the first three PCs were used for visualization.

Since each trial had a different latency to the cross event, trajectories were time-normalized. Each trial was divided into two segments: a fixed-length pre-open segment (Segment A, from the start to the open event; 55 s) and a variable-length post-open segment (Segment B, from the open event to the cross event). For trials without a defined Cross event (e.g., CS-only group), the average cross latency from the CS + Open group was used. Segment B was interpolated to a fixed length of 30 bins using shape-preserving piecewise cubic interpolation. The two segments were concatenated and smoothed using a moving average. The smoothing window size was determined heuristically to attenuate approximately 25% of the energy of the PCA scores for each trial.

Time-normalized trajectories were averaged within each group to produce group-level mean trajectories and corresponding standard errors of the mean (SEM). These trajectories, uniform in the number of time bins, were visualized either separately for each group or overlaid for direct comparison.

For both the inhibition simulation and the 180-s interval simulation, corresponding trial data were combined with CS + Open trials. These combined datasets were reshaped and processed using the same PCA and time-normalization procedures described above.

## Prediction of CS + Open trajectories from component conditions

To evaluate how well CS + Open trajectories could be reconstructed from their component conditions, we extracted time-normalized PCA trajectories from the CS + Open, Open-only, and CS-only groups. For the CS + Open condition, trajectories were aligned to the door-opening event, which occurred 5 s after CS offset; for the CS-only condition, trajectories were taken beginning at the corresponding time point—5 s after CS offset. All trajectories were preprocessed using the same procedures described in the trajectory visualization section. To standardize across conditions, trajectories were trimmed to include only the period from the open event to the cross event (30 bins), which served as the input for modeling.

We trained two models to predict individual CS + Open trial trajectories from the group-averaged Open-only and CS-only trajectories:

## Linear regression model

At each time bin, the trial trajectory was regressed onto the corresponding bins of the mean Open-only and CS-only trajectories, yielding a bin-wise linear estimate.

## Multilayer perceptron

To test for nonlinear integration, we implemented a feedforward neural network with two hidden layers ([6, 3] units) and L2 regularization ($\lambda \in [0, 0.25, 0.5, 0.75, 1]$), with the optimal $\lambda$ chosen to minimize the RSS for each condition. For each condition (CS + Open, inhibition, 180 s interval), the $\lambda$ yielding the lowest residual sum of squares (RSS) was selected. The MLP received concatenated mean trajectories from the Open-only and CS-only conditions as inputs and was trained to predict the trial trajectory of the CS + Open condition. Training was performed independently for each trial using a 70/15/15% split of bins for training, validation, and test sets, with min–max normalization applied to the inputs.

Prediction performance was quantified as the RSS at each time bin. For the MLP, training was repeated 10 times per trial, and RSS values were averaged across repetitions to yield a trial-wise mean RSS time series. To assess the benefit of nonlinear integration, improvement by the MLP over the linear model was calculated as the bin-wise difference between their RSS values, normalized by the linear model RSS. These normalized scores were averaged across bins to obtain a single summary value for each trial.

## Decoding analysis (for Figure 5A, B)

### CS decoding

We first gathered the time-normalized $Z$-scores of all cluster neurons from two trial types: CS + Open (cue present) and Open-only (cue absent). Time normalization from open to crossing was identical to the procedure described above.

For each time bin we formed a trial-by-neuron matrix and trained a linear discriminant model with fitcdiscr('linear') in MATLAB R2022b. Trials in which a cue was delivered were labeled 1 from cue onset onward, whereas cue-absent trials were labeled 0 throughout. Model performance was assessed by fivefold cross-validation, and accuracy was reported as 1 – kfoldLoss. Repeating this procedure over all bins produced a temporal profile of CS-decoding accuracy.

### Door open decoding

A Door-open decoder was generated in the same way, but the training data comprised CS + Open vs CS-only trials. Here, bins occurring after the outlet opened were labeled 1, and all earlier bins (as well as every bin in CS-only trials) were labeled 0. Fivefold cross-validated linear discriminant analysis again yielded a time-resolved accuracy curve that reflected when RNN activity best predicted door opening.

## PID analysis

To quantify how CS and door-open information were distributed across RNN neurons, we applied PID analysis to decoding accuracies as X variables and neural activity as a $Y$ in Clusters 1–3. For each cluster, $Z$-scored activity was extracted and time-normalized using a fixed window of 85 bins, which included a 55-s pre-open segment and a 30-bin post-open segment interpolated to a fixed length. Trials from four conditions were included: CS + Open, Open-only, CS-only, and None.

For each neuron and each time bin, we computed PID terms (redundancy, CS-unique information, Open-unique information, and synergy) using MATLAB code(the Neuroscience Information Theory Toolbox *Timme and Lapish, 2018*).

At each time bin, we used the neuron's $Z$-scored activity rounded to one decimal place across all trials as the $Y$ (time × trial), and the CS and door-open decoding scores as $X_1$ and $X_2$, respectively (time × trial) (see *Figure 5D*).

Specifically, the construction of $X_1$ (cue-related events) was as follows.

For the CS + Open and CS-only conditions, decoding accuracy were computed as described above. Trials in which a cue was delivered were labeled 1 from cue onset onward, whereas cue-absent trials were labeled 0 throughout. A decoder was trained to classify the $Z$-scored firing rates in each time bin of CS + Open and Open-only trials according to these labels. To express decoding accuracy as a

score ranging from 0 to 1, we subtracted 0.5 from the decoding accuracy, multiplied the result by 2, and set any negative values to 0. The resulting scores were smoothed with a Gaussian kernel (window size = 15). To compute the probability of coincident events, smoothed scores were rounded to the first decimal place. For Open-only and None conditions, all time bins were assigned a value of 0.

The construction of $X_2$ (door-open-related events) was as follows.

For CS + Open trials, as described above, time bins after the outlet opening were labeled 1, and all earlier bins as well as bins in CS-only trials were labeled 0. A decoder was trained to classify each time bin's $Z$-scored activity in CS + Open and CS-only trials based on these labels. Decoding scores were then computed using the same method as described for $X_1$.

For Open-only trials, time bins following outlet opening were labeled 1, and all earlier bins as well as bins in None trials were labeled 0. A decoder was trained to classify $Z$-score in Open-only and None trials, and decoding scores were computed identically.

For CS-only and None conditions, all bins were assigned a value of 0.

For each discrete state $j$ of $Y$ and each combination of states k from a source A (either $X_1$ alone, $X_2$ alone, or their joint distribution [$X_1$, $X_2$]), we computed the corresponding conditional probabilities required for PID analysis.

$$P(A = k|Y = j) = P(Y = j, A = k)/P(Y = j), P\left(Y = j|A = k\right) = P\left(Y = j, A = k\right)/P\left(A = k\right)$$

The information that source A conveys about Y when Y is in state j is then

$$I\left(Y = j; A\right) = \sum_k P\left(Y = j, A = k\right) \log_2 \frac{P\left(Y = j \mid A = k\right)}{P\left(Y = j\right)}$$

We defined its redundancy by, for each $j$, taking the minimum of $I\left(Y = j; A\right)$ over the $X_1$ and $X_2$, and then computing the dot product of that vector of minima with the state-probabilities $P\left(Y = j\right)$. $X_1$-unique information is the mutual information between $Y$ and $X_1$ minus the redundancy. $X_2$-unique information is the mutual information between $Y$ and $X_2$ minus the redundancy. Synergy is obtained by taking the joint mutual information, subtracting both unique-information terms, and then adding back the redundancy.

PID terms were averaged across neurons within each cluster. For visualization, time-resolved PID terms were shown as mean ± SEM across neurons.

## Cross-temporal decoding analysis

Cross-temporal decoding of CS + Open information was performed as follows: For each time bin, a decoder was trained to classify $Z$-scored neural activity from CS + Open trials (all time bins labeled as 1) vs CS-only and Open-only trials (all time bins labeled as 0). The trained decoder was then tested across all time bins to predict trial type. Decoding accuracy was computed as (correct trials/total trials × 100) and visualized as a two-dimensional heatmap indexed by training and testing time bins.

## Fixed point analysis

We implemented the FixedPointFinder toolbox to identify fixed points in the trained RNN model (*Golub and Sussillo, 2018*).

Fixed point analysis was performed separately for the following input epochs, defined by specific timestep intervals and corresponding input vectors:

Baseline (40–69 s): [0, 0, 0, 1, 0, 0, 0, 0]
CS (70–89 s): [2, 0, 0, 1, 0, 0, 0, 0]
Interval (90–94 s): [0, 0, 0, 1, 0, 0, 0, 0]
Open (95–239 s): [0, 1, 0, 1, 0, 0, 0, 0]

The recurrent and input weights, bias vector, and the alpha parameter ($\alpha$ = $\Delta t / \tau$) were used to represent the trained dynamics throughout the epochs.

For each epoch, the neural state trajectories $x$ and corresponding input vector $u$ were constructed. Initial conditions for fixed point searches were formed by combining all neural states within the epoch with a randomly sampled subset of global network states ($n$_trial = (number of epoch states) + 512

randomly sampled states; *x*: *n*_trial × *n*_neurons matrix,*u*:*n*_trial × 8 matrix). The input vector *u* was tiled across trials to match the dimensionality.

Fixed points were then identified jointly via optimization To find fixed point candidates, *x* and their one-step forward states *F* were computed, and the difference between *x* and *F* was minimized in an iterative gradient descent loop, where *x* was updated using the Adam optimizer. If change in loss became smaller than a convergence tolerance($1 \times 10^{-4}$) times the learning rate or if the loss is smaller than tolerance, then the iteration was stopped. The maximum number of iterations was set to 10,000. We discarded the candidate as an outlier if the distance between a fixed point candidate *x\** and the centroid of the initial states was larger than 10.0 times the average distance between the centroid and each initial state. Redundant candidates were further merged if the Euclidean distance between any pair of *x\** was less than $1 \times 10^{-2}$.The remaining fixed points *x\** and their Jacobian eigenvalues were collected. Fixed points were classified based on their dynamical stability: points were labeled as stable (attractors) if all eigenvalues had absolute magnitudes less than 1.0, and as saddle points otherwise.

## Trajectory analysis of biological data using GPFA

For the biological data, *Z*-scored activity pooled from in vivo recordings of nonexploratory Cluster 1 neurons was reduced to a three-dimensional space using GPFA, with Gaussian smoothing kernel size of 20 bins (implemented using MATLAB code from *Lakshmanan et al., 2015*). Each pseudopopulation trajectory from CS or Neutral rats was subsequently smoothed with a Gaussian kernel (10-bin window) and visualized. We employed GPFA because it estimates latent states shared across neurons while explicitly modeling the noise characteristics of individual neurons, making it well-suited for single-trial trajectory visualization. Unlike PCA, which identifies orthogonal axes that explain maximal variance without incorporating temporal structure, GPFA accounts for temporal dynamics, providing a key advantage in analyzing population activity over time.

Validation was performed by independently applying GPFA to two neuronal subsets derived from in vivo CS group recordings. Neurons were ranked in descending order based on their mean *Z*-scored firing rates during the baseline to 5-s post-crossing window. The ranked list was then divided into two subsets comprising odd- and even-indexed neurons, respectively. GPFA was applied to each subset using identical parameter settings as described above. For visualization, neural trajectories were smoothed with a 10-bin Gaussian window. The trajectory of one subset (subset 1) was chosen as the reference, and the others were aligned to it using the Procrustes method (MATLAB function). Specifically, anchor points from each trajectory were mapped to the corresponding anchors in the reference trajectory by minimizing the sum of squared errors under a linear transformation. The procedure included centering and normalization, followed by singular value decomposition to estimate the optimal rotation, and finally applying translation and scaling. Anchor points consisted of the trial start, CS onset, CS offset, door opening, and crossing time.

## Acknowledgements

This work was supported by the NRF (https://www.nrf.re.kr) of Korea grant funded by the Korean Government Ministry of Education (https://english.moe.go.kr), Science and Technology (NRF-2021R1A2B5B03002345), awarded to SC.

## Additional information

### Funding

| Funder | Grant reference number | Author |
|---|---|---|
| National Research Foundation of Korea | NRF-2021R1A2B5B03002345 | Sukwoo Choi |

The funders had no role in study design, data collection, and interpretation, or the decision to submit the work for publication.

## Author contributions
Kuenbae Sohn, Donghyeon Yoon, Conceptualization, Data curation, Software, Formal analysis, Validation, Investigation, Methodology, Writing – original draft, Writing – review and editing; Junghwa Lee, Conceptualization, Data curation, Formal analysis, Supervision, Validation, Investigation, Methodology, Writing – original draft, Project administration, Writing – review and editing; Sukwoo Choi, Conceptualization, Resources, Supervision, Funding acquisition, Validation, Investigation, Methodology, Writing – original draft, Project administration, Writing – review and editing

## Author ORCIDs
Kuenbae Sohn ⓘD https://orcid.org/0000-0001-6301-6403
Donghyeon Yoon ⓘD https://orcid.org/0000-0002-2173-7832
Junghwa Lee ⓘD https://orcid.org/0000-0001-5656-4399
Sukwoo Choi ⓘD https://orcid.org/0000-0002-6445-4912

## Ethics
All animal experiments were approved by the Institutional Animal Care and Use Committee (IACUC) of Seoul National University (approval No. SNU-210122-2-3; approved on July 21, 2022) and were performed in accordance with the guidelines for the care and use of laboratory animals of Seoul National University. All surgical procedures were performed under appropriate anesthesia, and all efforts were made to minimize animal suffering.

Reviewer #1 (Public review): https://doi.org/10.7554/eLife.109539.3.sa1
Reviewer #2 (Public review): https://doi.org/10.7554/eLife.109539.3.sa2
Author response https://doi.org/10.7554/eLife.109539.3.sa3

# Additional files

## Supplementary files
MDAR checklist

## Data availability
All custom code and trained models used for RNN simulations and downstream analyses in this study are publicly available at GitHub: https://github.com/Kuenbae/claustrum_computation (copy archived at *Sohn and Yoon, 2026*).

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
