## [Editor Report · eLife Assessment]

This work provides an **important** modeling-based framework for understanding the processes of temporal integration in the claustrum. These mechanisms could support a broader range of integrative brain function. The manuscript presents **solid** evidence for how claustrum may integrate temporal disparate signals via a novel computational phenomenon with neural dynamics evolving along neural trajectories as opposed to settling into fixed-point attractor states.

---

## [Referee Report · Reviewer #1 (Public review)]

Summary:

In this manuscript, the authors investigate how the anterior claustrum may integrate temporally separated task-relevant signals to guide behavior in a delayed escape paradigm. Because in vivo neural recordings from claustrum during this task are extremely limited-comprising single-trial data with small neuronal samples-the authors adopt a modeling-driven approach. They train recurrent neural networks (RNNs) using only behavioral data (escape latency) to reproduce task performance and then analyze the internal dynamics of the trained networks. Within these networks, they identify a subset of units whose activity exhibits persistent responses and strong correlations with behavior, which the authors label as "claustrum-like." Using dimensionality reduction, decoding, and information-theoretic analyses, they argue that these units dynamically integrate conditioned stimulus (CS) and door-opening signals via nonlinear, trajectory-based population dynamics rather than fixed-point attractor states.

To bridge model predictions and biology, the authors complement the modeling with in vitro slice experiments demonstrating recurrent excitatory connectivity and prolonged activity in the anterior claustrum that depends on glutamatergic transmission. They further compare latent neural trajectories derived from previously published in vivo claustrum recordings to those observed in the RNN, reporting qualitative similarities. Based on these results, the authors propose that the claustrum implements temporal signal integration through recurrent excitatory circuitry and dynamic population trajectories, potentially supporting broader theories of integrative brain function.

Strengths:

This study addresses an important and challenging problem: how to infer population-level computation in a brain structure for which in vivo data are sparse and experimentally constrained. The authors are commendably transparent about these limitations and seek to overcome them through a principled modeling framework. The integration of behavioral modeling, RNN analysis, and slice electrophysiology is ambitious and technically sophisticated.

Several aspects stand out as strengths. First, the behavioral RNN is carefully trained and interrogated using a rich set of modern analytical tools, including cross-temporal decoding, trajectory analysis, and partial information decomposition, providing multiple complementary views of network dynamics. Second, the slice experiments convincingly demonstrate recurrent excitatory connectivity in anterior claustrum, lending biological plausibility to the model's reliance on recurrent dynamics. Third, the manuscript is clearly written, logically organized, and conceptually engaging, and it offers a coherent mechanistic hypothesis that could guide future large-scale recording experiments.

Importantly, the work has significant heuristic value: rather than merely fitting data, it attempts to generate testable computational ideas about claustral function in a regime where direct empirical access is currently limited.

Weaknesses:

Despite these strengths, the manuscript suffers from a recurring and substantial conceptual issue: systematic over-interpretation of model-data correspondence. While the modeling results are potentially insightful, the extent to which they are presented as recapitulating real claustral neural mechanisms goes beyond what the available data can support.

A fundamental limitation is that the RNN is trained solely on behavioral output, without being constrained by neural data at either single-unit or population levels. As a result, the internal network dynamics are underdetermined and non-unique. Many distinct internal solutions could plausibly generate identical behavior. However, the manuscript frequently treats the specific internal solution discovered in the RNN as if it were a close approximation of the actual claustrum circuit.

This issue is compounded by the sparse nature of the in vivo data used for comparison. The GPFA-based trajectory analyses rely on pseudo-populations and single-trial recordings, yet are interpreted as evidence for robust population-level dynamics. Because neurons were not recorded simultaneously, the inferred trajectories necessarily lack true population covariance and shared trial-to-trial variability, limiting their interpretability as genuine population dynamics. Similarly, conclusions about trajectory-based versus attractor-based computation are drawn almost exclusively from model analyses and then generalized to the biological system.

Overall, while the modeling framework is appropriate as a hypothesis-generating tool, the manuscript repeatedly crosses the line from proposing plausible mechanisms to asserting explanatory or even causal equivalence between the model and the brain. This undermines the otherwise strong contributions of the work.

Below are several specific points that warrant further clarification or revision:

(1) Tone of model-data correspondence

Numerous statements describe the RNN as "closely mimicking," "recapitulating," or being "nearly identical" to claustral neural dynamics, sometimes extending to claims about causal relationships between neural activity and behavior. Given that neural data were not used to train the model, and that only a small subset of trained networks showed the reported dynamics, these statements should be substantially softened throughout the manuscript. The RNN should be framed as providing one possible computational realization consistent with existing data, not as a close instantiation of the biological circuit.

(2) Non-uniqueness of RNN solutions

The fact that only a small fraction of trained networks exhibited "claustrum-like" clusters deserves deeper discussion. This observation raises the possibility that the identified solution is fragile or highly specific rather than canonical. The authors should explicitly discuss the non-uniqueness of internal solutions in behavior-trained RNNs, including the range of alternative network dynamics that can reproduce the same behavior. In particular, it should be clarified why the specific network exhibiting "claustrum-like" clusters is informative about claustral computation, rather than representing one arbitrary solution among many.

(3) GPFA trajectory comparisons

The qualitative similarity between RNN trajectories and GPFA-derived trajectories from sparse in vivo data is interesting but insufficient to support claims of robustness or population-level structure. Statements suggesting that these patterns are unlikely to arise from noise or random fluctuations are not justified given the single-trial, pseudo-population nature of the data. Either additional quantitative controls should be added, or the interpretation should be substantially tempered.

(4) Scope of functional claims

The discussion connecting the findings to broad theories of claustral function, global workspace, or consciousness extends well beyond the data presented. These speculative links should be clearly labeled as such and significantly reduced in strength and prominence.

The manuscript repeatedly describes the delayed escape task as an "inference-based behavioral paradigm" and states that animals "infer that a value-neutral alternative space is likely to be safer" when the CS is presented in a novel environment. While I appreciate that the US-CS association was established in a different context and that the CS is then presented in a new environment, I am not convinced that the current behavioral evidence uniquely supports an inference interpretation.

First, it is not clear that this task is widely recognized in the literature as a canonical inference task, in the sense of, for example, sensory preconditioning, transitive inference, or model-based inference paradigms. Rather, the observed effect-that CS animals escape faster to a neutral compartment than neutral-CS controls-can be parsimoniously interpreted in terms of generalized threat value, heightened fear/anxiety, or a bias toward avoidance/escape under elevated threat, without requiring an explicit inferential step about the specific safety of the alternative compartment. The fact that no prior training is needed is compatible with flexible generalization, but does not by itself demonstrate inference in a more formal computational sense.

Second, the inference claim becomes central to the manuscript's conceptual framing (e.g., the idea that rsCla supports "inference-based escape"), yet the behavioral analyses presented here and in the cited prior work do not clearly rule out simpler accounts. Clarifying this distinction would help avoid overstating both the inferential nature of the behavior and the specific role of rsCla and the RNN's "claustrum-like" cluster in supporting inference per se, as opposed to more general integration of threat-related signals with an opportunity for escape.

This manuscript presents an interesting and potentially valuable modeling-based framework for thinking about temporal integration in the claustrum, supported by solid slice physiology. However, in its current form, it overstates the degree to which the proposed RNN dynamics reflect actual claustral neural mechanisms. With substantial revision-especially a more cautious interpretation of model-data similarity and a clearer articulation of modeling limitations-the study could make a meaningful contribution as a hypothesis-generating work rather than a definitive mechanistic account.

Comments on revisions:

The authors have carefully addressed the concerns raised in the initial review. In particular, the manuscript has been substantially improved in terms of tone, conceptual clarity, and the interpretation of the modeling results. The revised version now presents a well-balanced and appropriately framed account of the work.

The study offers a compelling and useful hypothesis-generating framework for understanding temporal integration in the claustrum, and I support its publication. As a minor point, given the acknowledged limitations of pseudo-population and single-trial data, it would be preferable to slightly soften a few remaining statements that describe trajectory structure as directly "reflecting" population-level dynamics (e.g., using "consistent with" instead).

---

## [Referee Report · Reviewer #2 (Public review)]

This manuscript reports the behavior of a computational model of rat claustral neurons during the performance of a behavioral task known as the delayed escape task (in this reviewer's understanding, this behavioral task was created and implemented by this group only). These authors have argued in a prior manuscript (Han et al.) that a group of neurons located "rostral to striatum" are part of the claustrum. The group names the region the "rostral to striatum claustrum." Additionally, in the Han et al. paper, the authors argue that these cells are responsible for maintaining a signal that lasts through the delay period.

The main findings of the current paper are:

(1) The authors have built a model network that was trained to show firing similar to what was reported for rats in their prior paper.

(2) The authors' analysis of model behavior is used to suggest that the model network recapitulates biological activity, including the existence of a cluster of cells mainly responsible for the delay period firing.

(3) The authors offer evidence from patch clamp recordings for excitatory interconnections among claustral neurons that are an essential feature of the model network.

A major value of the computational network is that "trials" of the network can be performed. In experiments on animals, only single trials can be used.

Concerns:

(1) This paper is based on behavioral results and neural recordings from their prior paper (Han et al.), but data, e.g. in figure 1, are not clearly identified as new or as coming from that source. Figure 1A, for example, appears to be taken directly from Han et al. No methods are given in this manuscript for the behavioral testing or the in vivo electrophysiology.

(2) Many other details are unclear. Examples include model training, the weight matrices and how these changed with training (p. 13), the equations 2 and 3 (p. 13), the sources for the constants in the equations (p. 14), the methods (anesthesia, stereotaxic coordinates, injection specifics and details for "sparse expression") for the ChrimsonR injections.

(3) The explorations of model behavior are a catalog of everything tried rather than an organized demonstration of what the model can and cannot do. The figures could be reduced in number to emphasize the key comparisons of the different clusters and the model's behavior under different conditions intended to "test" the model.

(4) On page 6, the E-E connectivity is argued from Shelton et al. (2025) and against Kim et al. (2016), but ignores Orman (2015), which to this reviewer's knowledge was the first to demonstrate such connectivity, including the long duration events and impact of planes of section.

(5) Whereas the authors are entitled to their own opinion of prior work (references 3-8), it is inappropriate to misrepresent prior work as only demonstrating a "limited function" of claustum. Additional papers by Mathur's group and Citri's group are ignored.

In summary, the authors have made a computational model that recapitulates the firing of a subset of potentially claustral neurons during a particular behavioral task (delayed escape is certainly not the only behavior that involves claustrum - see e.g., attention, salience, sleep). If the conclusion is that excitatory claustral cells must be connected to other excitatory claustral cells, such a conclusion is not new and the electrophysiological E-E metrics are not well quantified (e.g., connectivity frequency, strength of connection). If the model is intended to predict how claustrum might accomplish any other task, there is insufficient detail to evaluate the model beyond the evidence that the model creates a subset of cells that can sustain firing during the delay period in the delayed escape task.

All relevant work must be appropriately cited throughout the manuscript.

Comments on revisions:

The authors have adequately addressed the concerns that were raised in response to the first version of the manuscript.

---

## [Author Response]

The following is the authors’ response to the original reviews

**Public Reviews:**

**Reviewer #1 (Public review):**

We thank the reviewer for their constructive and insightful comments and agree with the importance of the points raised. We recognize that aspects of our original presentation may have been unclear or overly strong in their interpretation. We have therefore revised the manuscript to clarify our intended scope, moderate our claims, and strengthen the analysis. In the second paragraph of the Discussion, we have explicitly acknowledged the concerns raised by the reviewer and outlined how they have been addressed in the revised manuscript. Our detailed responses are provided below.

(1) Tone of model-data correspondenceNumerous statements describe the RNN as "closely mimicking," "recapitulating," or being "nearly identical" to claustral neural dynamics, sometimes extending to claims about causal relationships between neural activity and behavior. Given that neural data were not used to train the model, and that only a small subset of trained networks showed the reported dynamics, these statements should be substantially softened throughout the manuscript. The RNN should be framed as providing one possible computational realization consistent with existing data, not as a close instantiation of the biological circuit

We agree with the reviewer’s comment. The expressions noted by the reviewer (e.g., closely mimicked, nearly identical, recapitulate) will be replaced with alternative wording that conveys a more moderate meaning (Line 16-17, 65-66, 83, 96, 120, 212).

(2) Non-uniqueness of RNN solutionsThe fact that only a small fraction of trained networks exhibited "claustrum-like" clusters deserves deeper discussion. This observation raises the possibility that the identified solution is fragile or highly specific rather than canonical. The authors should explicitly discuss the non-uniqueness of internal solutions in behavior-trained RNNs, including the range of alternative network dynamics that can reproduce the same behavior. In particular, it should be clarified why the specific network exhibiting "claustrum-like" clusters is informative about claustral computation, rather than representing one arbitrary solution among many.

As the reviewer pointed out, behaviorally trained RNNs can admit multiple internal solutions that produce the same behavioral output, and we acknowledge the non-uniqueness of such internal solutions. However, we do not interpret the fact that only a subset of trained RNNs exhibit dynamics similar to those observed in the claustrum as evidence that this solution is fragile. Notably, the claustrum-like dynamics emerged spontaneously during training and were not explicitly enforced. Furthermore, our finding suggests that the emergence of this particular dynamical regime depends on relatively specific structural constraints.

Our criterion for selecting RNNs that could inform the computational principles of the claustrum was their ability to reproduce the behavioral and physiological observations obtained in the delayed escape experiments. RNNs that were excluded may reflect information-processing strategies used by other brain regions or may rely on artificial logical structures. The computational demand of the task, which integrates temporally separated signals, naturally drives convergence toward networks with recurrent excitatory connectivity capable of maintaining persistent activity. Indeed, all networks that exhibited a claustrum-like cluster shared a common structural feature: strong recurrent excitatory connectivity within Cluster 1. This property is consistent with biological characteristics observed in the slice experiments shown in Fig 2.

Importantly, the computational principles derived from this RNN were found to be quantitatively consistent with in vivo single-neuron activity patterns. Specifically, analysis using an eigenvalue-based metric (λ_3_/Σλ) revealed the same directional effect in both the RNN and the claustrum neuron data. In addition, a leave-one-neuron-out analysis showed that this pattern was broadly distributed across in vivo claustral neurons rather than being driven by a small subset (see Fig. 4).

Taken together, these convergent lines of evidence suggest that the computational model is not simply one arbitrary solution among many possible alternatives, but rather implements a computational principle that may underlie claustral functions.

(3) GPFA trajectory comparisonsThe qualitative similarity between RNN trajectories and GPFA-derived trajectories from sparse in vivo data is interesting but insufficient to support claims of robustness or population-level structure. Statements suggesting that these patterns are unlikely to arise from noise or random fluctuations are not justified, given the single-trial, pseudo-population nature of the data. Either additional quantitative controls should be added, or the interpretation should be substantially tempered.

As the reviewer pointed out, the GPFA trajectory comparison presented in the original manuscript remained largely qualitative, and we agree that this alone was insufficient to establish robustness or provide convincing evidence for population-level structure. In the revised manuscript, we have therefore added the requested quantitative analysis (see Fig. 4).

Before describing the analysis, we would like to clarify several methodological limitations associated with pseudopopulation and single-trial data. GPFA estimates latent trajectories based on assumptions about covariance structure among neurons and temporal smoothness. In pseudopopulation datasets, the true simultaneously recorded covariance structure cannot be fully reconstructed, which is an inherent limitation. Because our dataset is based on single trials, the analysis does not directly exploit trial-to-trial variability. Nevertheless, the estimation of the latent space still depends on the covariance structure among real claustral neurons, suggesting that the inferred trajectories remain tied to biologically meaningful population dynamics.

Accordingly, the quantitative metric we introduce is not entirely independent of the GPFA estimation step. Rather, it is intended to evaluate the geometric structure of the single-trial latent trajectories estimated by GPFA. We acknowledged this limitation in the revised manuscript.

Specifically, for the biological data, we reanalyzed the GPFA-derived latent trajectories in PCA space and computed an eigenvalue-based metric (λ_3_/Σλ). For each of the 20 time bins, we applied a sliding window of 10 bins and calculated the covariance matrix within that window. The eigenvalues of PC1, PC2, and PC3 were then obtained, and the third eigenvalue (λ_3_) was normalized by the total variance (Σλ = λ_1_ + λ_2_ + λ_3_). This metric quantifies the degree to which the trajectory locally deviates from a planar structure that can be explained by two dominant axes. An increase in λ_3_/Σλ indicates that the population-state trajectory forms a higher-dimensional geometric structure beyond a simple two-dimensional combination.

For the RNN data, in contrast, the activity of all units can be observed simultaneously and sufficient trial repetitions are available. Therefore, GPFA was not applied; instead, PCA was performed directly on the population activity for each trial. We then computed an average trajectory across trials and applied the same λ_3_/Σλ metric. Thus, although the initial dimensionality reduction steps differ between the two systems, the definition and calculation of the final quantitative metric are identical. The focus of the comparison is therefore not the dimensionality reduction technique itself, but the geometric dimensional structure of the population trajectories evolving over time.

Importantly, within the biological dataset, the GPFA estimation procedure, preprocessing steps, pseudopopulation construction, subsampling strategy, temporal alignment criteria, and smoothing parameters were applied identically across conditions. Likewise, the same analysis pipeline was used for all conditions in the RNN. If structural biases had been introduced during covariance estimation or dimensionality reduction, they would be expected to affect all conditions within each system similarly. Nevertheless, the λ_3_/Σλ value was consistently and significantly higher in the CS condition than in the Neutral condition, and this directional pattern was observed in both the RNN and the claustral neuron data. This suggests that the effect reflects condition-specific differences in population dynamical structure rather than artifacts arising from a particular dimensionality reduction method.

To further test whether the observed effect might be driven by a small subset of neurons or specific neuron combinations, we performed a leave-one-neuron-out analysis on the claustrum dataset. Recomputing λ_3_/Σλ while removing one neuron at a time showed that, in the CS group, most neurons contributed relatively evenly to this metric, whereas the Neutral group did not show such a distributed contribution pattern. This indicates that the observed three-dimensional structure is not driven by a few outlier neurons or incidental covariance patterns, but rather reflects an organized population-level phenomenon.

If the result were primarily due to structural artifacts introduced by the pseudopopulation construction or dimensionality reduction procedures, it would be unlikely for consistent selective differences to repeatedly emerge between conditions under identical analysis pipelines. The consistently higher λ_3_/Σλ values observed in the CS condition therefore provide indirect support that this pattern reflects condition-specific population dynamics rather than estimation bias.

Taken together, these results suggest that the observed three-dimensional structure reflects condition-specific population dynamics rather than analysis artifacts. The fact that the same quantitative metric yields consistent effects in both the RNN and claustral data further strengthens the correspondence between the two systems.

(4) Scope of functional claimsThe discussion connecting the findings to broad theories of claustral function, global workspace, or consciousness extends well beyond the data presented. These speculative links should be clearly labeled as such and significantly reduced in strength and prominence.

We agree with the reviewer and stated that references to these theories are speculative, while substantially reducing both their emphasis and prominence in the manuscript (Line 444-446, 451).

(5) Comment on Conceptual Interpretation of the Behavioral Paradigm:The manuscript repeatedly describes the delayed escape task as an "inference-based behavioral paradigm" and states that animals "infer that a value-neutral alternative space is likely to be safer" when the CS is presented in a novel environment. While I appreciate that the US-CS association was established in a different context and that the CS is then presented in a new environment, I am not convinced that the current behavioral evidence uniquely supports an inference interpretation.First, it is not clear that this task is widely recognized in the literature as a canonical inference task, in the sense of, for example, sensory preconditioning, transitive inference, or model-based inference paradigms. Rather, the observed effect-that CS animals escape faster to a neutral compartment than neutral-CS controls-can be parsimoniously interpreted in terms of generalized threat value, heightened fear/anxiety, or a bias toward avoidance/escape under elevated threat, without requiring an explicit inferential step about the specific safety of the alternative compartment. The fact that no prior training is needed is compatible with flexible generalization, but does not by itself demonstrate inference in a more formal computational sense.Second, the inference claim becomes central to the manuscript's conceptual framing (e.g., the idea that rsCla supports "inference-based escape"), yet the behavioral analyses presented here and in the cited prior work do not clearly rule out simpler accounts. Clarifying this distinction would help avoid overstating both the inferential nature of the behavior and the specific role of rsCla and the RNN's "claustrum-like" cluster in supporting inference per se, as opposed to more general integration of threat-related signals with an opportunity for escape.

We agree with the reviewer’s concern. First, we referred to the delayed escape behavioral task as “a behavioral paradigm that requires integration of temporally separated task-relevant signals.” (Line 7-8). We also removed references to the term inference throughout the manuscript (Line 46, 51, 67, 397).

**Reviewer #2 (Public review):**

We sincerely thank the reviewer for their constructive and insightful comments. Through the revision process, the manuscript has been substantially improved, with increased reproducibility, more appropriate acknowledgment of prior work, and a clearer and more logical presentation of the study.

(1) This paper is based on behavioral results and neural recordings from their prior paper (Han et al.), but data, e.g., in Figure 1, are not clearly identified as new or as coming from that source. Figure 1A, for example, appears to be taken directly from Han et al. No methods are given in this manuscript for the behavioral testing or the in vivo electrophysiology.

We agree with the reviewer that this distinction should be made clearer. In the original manuscript, we indicated in the Figure 1 legend that panels A, D, E, F, and L (left) were reproduced from Han et al. (2024). To further clarify this point, we explicitly noted this distinction again in the main text (Line 74, 85). In addition, we described the behavioral experiments and in vivo electrophysiological recordings performed in Han et al. (2024) in the Methods section and include the appropriate citation (Line 463-530).

(2) Many other details are unclear. Examples include model training, the weight matrices and how these changed with training (p. 13), equations 2 and 3 (p. 13), the sources for the constants in the equations (p. 14), the methods (anesthesia, stereotaxic coordinates, injection specifics and details for "sparse expression") for the ChrimsonR injections.

We agree with the reviewer’s comment and have revised the manuscript to provide a more detailed description of the model training procedure, weight initialization, and parameter selection.

We expanded the explanation of the model training procedure and weight initialization. Specifically, the recurrent (W_rec_) and output (W_out_) weight matrices were initialized using a Glorot normal distribution with a standard deviation of \begin{document}$\sigma=\sqrt{2 /\left(N_{\text {in}}+N_{\mathrm{out}}\right)}$\end{document} to ensure stable signal propagation during early training. In addition, we now explicitly describe the training algorithm and optimization procedure. The network was trained using the Adam optimizer implemented in TensorFlow (v2.1.0) with a batch size of 256 for 1.2 million training iterations, minimizing the per-trial loss function defined in the manuscript. We also explicitly stated how Dale’s principle was maintained throughout training: rows in W_out corresponding to inhibitory units were zeroed out, and recurrent weights were continuously constrained so that excitatory and inhibitory neurons preserved their respective positive and negative synaptic projections. To illustrate how the weight structure evolved during training, we explicitly reference Figure 2A, which visualizes the final mean inter-cluster synaptic weights and highlights the strong recurrent connectivity that emerged within Cluster 1. Regarding Equations 2 and 3 and their constants, we clarified that the target escape times used to anchor the network were based on experimentally measured behavioral latencies (48.7 s for the CS-present condition and 111.3 s for the CS-absent condition). Furthermore, the regularization coefficients (λ = 0.01 and λ_FR_ = 0.95) were selected through a grid search procedure to maintain biologically plausible firing rates while preventing overfitting.

We detailed the surgical procedures that were previously omitted. This includes the specific anesthesia protocol (sodium pentobarbital, 50 mg/kg, i.p.), stereotaxic mounting, and the exact coordinates for the rsCla (AP +2.95, ML ±1.95, DV -3.85 mm). To define "sparse expression," we specified that the AAV was diluted 1:4 in sterile saline. Finally, we included the precise injection parameters: delivery at 20 nL/min via a pressure injection system, with the pipette left in place for 10 minutes post-infusion to ensure adequate diffusion. (Line 635, 636-639, 641-643). We have added these contents in the Methods section.

(3) The explorations of model behavior are a catalog of everything tried rather than an organized demonstration of what the model can and cannot do. The figures could be reduced in number to emphasize the key comparisons of the different clusters and the model's behavior under different conditions, intended to "test" the model.

We agree with the reviewer’s comment and have reorganized the figures to focus on the key results. Specifically, we separated the original figures so that they correspond to (1) Presentation of an RNN model consistent with the results of actual claustral recordings, (2) identification of dimensionality-reduced population activity patterns in the model, (3) comparison of these patterns with population activity patterns derived from recorded claustral neurons, (4) proposal of a nonlinear integration mechanism, and (5) the suggestion that such integration may be implemented through dynamic coding. Using this figure organization, we first identify RNN models trained on behavioral metrics whose dynamics are consistent with experimental claustral recordings. We then compare the dimensionality-reduced population activity patterns of these models with those derived from recorded claustral neurons to evaluate their biological plausibility. After selecting the models that satisfy this criterion, we perform further analyses that would be difficult to achieve using real neural recordings alone. These analyses ultimately allow us to propose dynamic coding exhibiting nonlinear integration as a plausible computational mechanism.

(4) On page 6, the E-E connectivity is argued from Shelton et al. (2025) and against Kim et al. (2016), but ignores Orman (2015), which, to this reviewer's knowledge, was the first to demonstrate such connectivity, including the long-duration events and impact of planes of section.

We agree with the reviewer’s suggestion and will include a reference to Orman (2015). We have clarified that neuronal activity can persist for extended periods and that such persistent activity has been observed in claustral slices prepared at a specific slicing angle (Line 144).

(5) Whereas the authors are entitled to their own opinion of prior work (references 3-8), it is inappropriate to misrepresent prior work as only demonstrating a "limited function" of claustrum. Additional papers by Mathur's group and Citri's group are ignored.

We agree with the reviewer’s comment and have revised the relevant sentences in the Introduction section. We also included and acknowledged the contributions of previous studies by the Mathur group and the Citri group by adding additional references to their works (Line 36, 429).

In summary, the authors have made a computational model that recapitulates the firing of a subset of potentially claustral neurons during a particular behavioral task (delayed escape is certainly not the only behavior that involves claustrum - see e.g., attention, salience, sleep). If the conclusion is that excitatory claustral cells must be connected to other excitatory claustral cells, such a conclusion is not new, and the electrophysiological E-E metrics are not well quantified (e.g., connectivity frequency, strength of connection). If the model is intended to predict how the claustrum might accomplish any other task, there is insufficient detail to evaluate the model beyond the evidence that the model creates a subset of cells that can sustain firing during the delay period in the delayed escape task.All relevant work must be appropriately cited throughout the manuscript.

Regarding the E–E metric, we obtained the following result. When including recordings in which the whole-cell recording could not be completed, optogenetically evoked responses were observed in 38 out of 43 patched cells. This suggests that approximately 90% of the cells receive intra-claustral excitatory input. However, the current dataset does not allow us to quantify the connection probability or the strength of these connections.

As the reviewer pointed out, the RNN developed in this study is specifically designed for the delayed escape task, and we do not intend to claim direct generalization to other proposed functions of the claustrum, such as attention, salience, or sleep. The goal of this study is to computationally characterize the temporal integration mechanism of the claustrum observed in this specific task. We have included this in the Discussion section. In the second paragraph of the Discussion, we have explicitly acknowledged the concerns raised by the reviewer and outlined how they have been addressed in the revised manuscript.